# InSAR time series analysis of seasonal surface displacement dynamics on the Tibetan Plateau

Eike Reinosch[1], Johannes Buckel[2], Jie Dong[3], Markus Gerke[1], Jussi Baade[4], Björn Riedel[1]

[1] Institute of Geodesy and Photogrammetry, Technische Universität Braunschweig, Braunschweig, Germany

[2] Institute of Geophysics and extraterrestrial Physics, Technische Universität Braunschweig, Braunschweig, Germany

[3] School of Remote Sensing and Information Engineering, Wuhan University, Wuhan, China

[4] Department of Geography, Friedrich-Schiller-Universität Jena, Jena, Germany

*Correspondence to:* Eike Reinosch (e.reinosch@tu-braunschweig.de)

**Abstract.** Climate change and the associated rise in air temperature have affected the Tibetan Plateau to a significantly stronger degree than the global average over the past decades. This has caused deglaciation, increased precipitation and permafrost degradation. The latter in particular is associated with increased slope instability and an increase in mass-wasting processes, which pose a danger to infrastructure in the vicinity. Interferometric Synthetic Aperture Radar (InSAR) analysis is well suited to study the displacement patterns driven by permafrost processes, as they are on the order of millimeters to decimeters. The Nyainqêntanglha range on the Tibetan Plateau lacks high vegetation and features relatively thin snow cover in winter, allowing for continuous monitoring of those displacements throughout the year. The short revisit time of the Sentinel-1 constellation further reduces the risk of temporal decorrelation, making it possible to produce surface displacement models with good spatial coverage. We created three different surface displacement models to study heave and subsidence in the valleys, seasonally accelerated sliding and linear creep on the slopes. Flat regions at Nam Co are mostly stable on a multiannual scale but some experience subsidence. We observe a clear cycle of heave and subsidence in the valleys, where freezing of the active layer followed by subsequent thawing cause a vertical oscillation of the ground of up to a few centimeters, especially near streams and other water bodies. Most slopes of the area are unstable, with velocities of 8 to 17 mm yr$^{-1}$. During the summer months surface displacements velocities more than double on most unstable slopes due to freeze-thaw processes driven by higher temperatures and increased precipitation. Specific landforms, most of which have been identified as either rock glaciers, protalus ramparts or frozen moraines, reach velocities of up to 18 cm yr$^{-1}$. Their movement

shows little seasonal variation but a linear pattern indicating that their displacement is
predominantly gravity-driven.

## 1 Introduction

Permafrost describes subsurface material with a temperature of 0 °C or lower for at least two
consecutive years (French, 2017). Permafrost is covered by the active layer, which freezes and thaws
seasonally (Shur et al., 2005). This causes frost heave and subsidence of wet ground on the order of
centimeters due to the volume change associated with the ice-water phase transition. The amplitude
of this heave and subsidence cycle is dependent on the water content of the active layer and the
material of the ground (Matsuoka et al., 2003). On permafrost slopes the freezing and thawing of the
active layer reduces slope stability (Zhang and Michalowski, 2015) and might create solifluction lobes
(Matsuoka et al., 2001). Further examples of creeping landforms associated with permafrost are rock
glaciers (Haeberli et al., 2006) and protalus ramparts (Whalley and Azizi, 2003).

The Tibetan Plateau (TP) has been the object of many studies focusing on climate change over the
past decades, especially since it has become known, that its temperature has risen significantly faster
than the global average with a rate of 0.25°C per decade (Yao et al., 2000). This issue is exacerbated
by the importance of the TP as a source of fresh water for large parts of greater Asia (Messerli et al.,
2004). The TP is often referred to as the "Third Pole", as it carries the largest volume of frozen fresh
water after the North- and South-Pole. The rising temperature has led to deglaciation at rates of over
0.2 % $yr^{-1}$ (Ye et al., 2017) and permafrost degradation (Wu et al., 2010) throughout the plateau,
increasing the river runoff by 5.5 % (Yao et al., 2007). Approximately 40 % of the TP is considered
permafrost and 56 % seasonally frozen ground (Zou et al., 2017). Permafrost is vulnerable to climate
change (Schuur et al., 2015) and it has been shown, that climate warming may accelerate permafrost
related creeping and sliding (Daanen et al., 2012). Glaciers and their retreat are very well
documented on the TP, as they can be assessed using optical satellite data with high accuracy (e.g.
Bolch et al., 2010). Permafrost features, such as rock glaciers or buried ice lenses, are harder to
quantify using optical remote sensing due to their relatively slow motion and often smaller size than
glaciers (Kääb, 2008). This has led to a severe lack of inventories documenting these permafrost
features, despite their importance as water storages (Jones et al., 2019) and the vulnerability of rock
glaciers to climate warming (Müller et al., 2016).

Permafrost related displacement processes, such as rockslides and the creeping of rock glaciers, can
be monitored through the collection of in-situ surface (e.g. Böhme et al., 2016) and subsurface data
(e.g. Kneisel et al., 2014) or with terrestrial remote sensing techniques like laser scanners (e.g. Bauer

et al., 2003).These techniques are generally labor intensive, require access to the often remote study sites and provide only sparse spatial coverage. Satellite-based remote sensing does not require access to the study sites and provides large spatial coverage, making it a valuable tool for the study of permafrost related displacements. The displacements vary from a few centimeters (heave and subsidence of the active layer) to decimeters or meters per year (creep of rock glaciers) and are therefore often too small to be studied with optical satellite techniques (Kääb, 2008). Cloud cover may inhibit the collection of continuous optical time series data (Joshi et al., 2016). However, satellites emitting microwaves, like the Sentinel-1 constellation launched by ESA in 2014, make the continuous detection of these displacements possible through Interferometric Synthetic Aperture Radar (InSAR) techniques. InSAR analysis is an active remote sensing technique, which exploits phase changes of backscattered microwaves to determine relative surface displacements taking place between two or more acquisition dates (Osmanoğlu et al., 2016). Other studies have employed InSAR techniques to study permafrost related processes on the TP (Li et al., 2015; Daout et al., 2017), north-western Bhutan (Dini et al., 2019), Norway (Eriksen et al. 2017), Svalbard (Rouyet et al., 2019) and Siberia (Antonova et al., 2018). Both seasonal processes, such as the heave and subsidence of freezing and thawing ground, and multiannual processes, like creep of periglacial landforms, have been studied. However, interpreting InSAR data can be challenging and often a number of assumptions have to be made. InSAR results provide only motion towards the satellite or away from it, not absolute ground displacement. It is therefore very difficult to accurately assess ground motion, without making assumptions about its actual direction. Unlike optical satellites, which observe the earth from a vertical Line-Of-Sight (LOS), SAR satellites are side-looking and observe the earth obliquely.

This paper presents the results of an analysis of ground movement in the permafrost prone area of the eastern and southern shores of Nam Co based on three to four year time series of Sentinel-1 acquisitions. We identify the various surface processes driving surface displacement around Nam Co on the southern TP and evaluate their potential causes. It is vital to understand these displacement patterns and to compare our results to similar studies, as the TP has been shown to react heterogeneously to climate change (Song et al., 2014). To that end we developed multiple surface displacement models, analyzing geomorphological processes in the valleys and on the mountain slopes on both a seasonal and a multiannual scale.

## 2 Study Area

The Nam Co is the second largest lake of the TP (Zhou et al., 2013), with a catchment covering an area of 10,789 km², 2018 km² of which is the lake's own surface area (Zhang et al., 2017). The proximity to Lhasa, its accessibility and the presence of the Nam Co Monitoring and Research Station for Multisphere Interactions CAS (NAMORS, Fig. 1), have made it a prime location to study the effects of climate change on the TP. The current lake level lies at 4726 m a.s.l. (Jiang et al., 2017) but it has featured a rising trend of approximately 0.3 m yr$^{-1}$ over the past decades (Kropáček et al., 2012; Lei et al., 2013). The eastern and southern borders of the catchment are defined by the Nyainqêntanglha mountain range with elevations of up to 7162 m a.s.l. The highest parts are glaciated (Bolch et al., 2010), while most other areas are considered to be in the periglacial zone (Keil et al., 2010; Li et al., 2014).

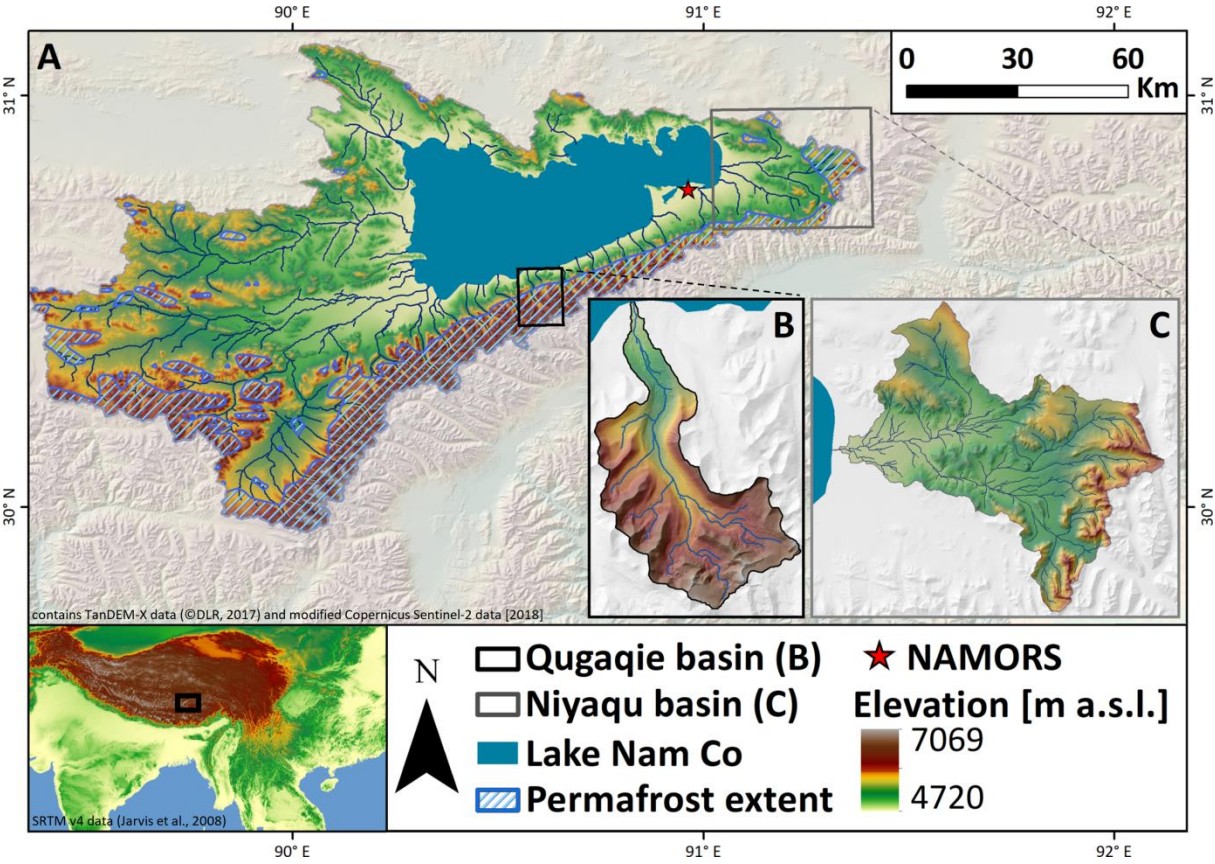

*Fig. 1:* *Overview map of the Nam Co catchment* ***(A)*** *including the locations of the NAMORS research station and the two main study areas: Qugaqie basin* ***(B)*** *and Niyaqu basin* ***(C)***. *Elevation data is based on SRTM v4 (Jarvis et al., 2008) and TanDEM-X 0.4" DEM (©DLR, 2017). Permafrost extent according to Zou et al. (2017) and lake extent based on the Normalized Difference Water Index (NDWI) of Sentinel-2 optical imagery (©Copernicus Sentinel data 2018, processed by ESA).*

The climate at the Nam Co is dominated by the Indian Monsoon in summer and the Westerlies in winter (Yao et al., 2013). The former brings warm moist air from the south, providing 250 to 450 mm of rainfall from June to September and accounting for approximately 80% of the annual precipitation (NAMORS, 2018; location in Fig. 1A). The Westerlies maintain semi-arid to arid conditions during the rest of the year. The snow cover is relatively sparse in winter, due to low precipitation outside of the monsoon season. The vegetation consists primarily of alpine steppe (Li, 2018), with high vegetation, such as shrubs and trees, being almost completely absent. The sparse snow cover and the lack of high vegetation make this region a prime study site for surface displacement related to periglacial processes using InSAR technology. Wang et al. (2017) used a combination of InSAR and optical satellite data to map rock glaciers in the northern Tien Shan of China, where the winters are similarly dry. The risk of temporal decorrelation, i.e. the loss of data coverage due to considerable change of physical surface characteristics, is lower than in other regions where such processes may be studied, such as Norway (Eriksen et al., 2017) or the Sierra Nevada in the USA (Liu et al., 2013). These regions feature considerable snow cover during long periods of the year, making continuous temporal coverage of fast-moving structures, like rock glaciers, difficult. This is especially a problem for satellites with the shorter X-band (2 - 4 cm) or C-band (4 - 8 cm) wavelengths, like TerraSAR-X (3.1 cm) and Sentinel-1 (5.6 cm), as they are more susceptible to temporal decorrelation (Crosetto et al., 2016) compared to systems with a longer wavelength such as L-Band (15 - 30 cm).

The two areas of interest for this study are the Qugaqie basin (58 km²) within the western Nyainqêntanglha mountain range, south of the Nam Co and the Niyaqu basin (409 km²) at the eastern Nyainqêntanglha mountain range, on the eastern shore of the lake (Fig. 1). These sub-catchments were chosen, as they feature different levels of glacial impact and represent the predominant landscapes and their related surface processes at Nam Co. The Niyaqu basin represents the majority of Nam Co's catchment with extensive alpine steppe vegetation and wetlands surrounded by hills with little exposed bedrock in the lower regions. The global permafrost map of Zou et al. (2017) shows, that permafrost is limited to the higher parts of the sub-catchment, at the eastern Nyainqêntanglha mountain range. The Qugaqie basin represents the periglacial landscape of the western Nyainqêntanglha mountain range. 60 % of its area are considered periglacial landforms (Li et al., 2014), some of which are still active in the higher parts of the catchment due to their potential ice content, such as rock glaciers. Rock glaciers are steadily creeping ice-rich debris on mountainous slopes associated with permafrost (Haeberli et al., 2006). Other landforms were shaped by fluvial, glacio-fluvial, glacial and aeolian processes (Keil et al., 2010). The vegetation cover is similar to the Niyaqu basin but with more areas of exposed glacial valley fill and bedrock interspersed in between the vegetated areas. Both the valleys and the slopes are covered by unconsolidated debris, mostly coarse gravel and boulders, and some slopes in the higher parts are free of soil and

vegetation. Steep topography and the presence of warming permafrost can be associated with rock slope instabilities, such as rock falls and rock slides (Fischer et al., 2006), making them a likely occurrence throughout Qugaqie basin and in the higher zones of Niyaqu basin. The bedrock consists of sandstone and carbonates in the lower areas of the basins and granodiorite and meta-sedimentary rocks in the higher parts (Kapp et al., 2005, Yu et al., 2019). The main river is fed by hanging valleys, some containing glaciers, as well as the two main glaciers Zhadang and Genpu to the south. The glaciers cover 8.4 % of the basin`s surface area and account for 15 % of its runoff in summer (Li et al., 2014). Two automated weather station and a rain gauge were operated near the ablation zone of the Zhadang Glacier between 2005 and 2010. Daily temperature averages range from approximately -15°C in winter to 3°C in summer in the Qugaqie basin and -10°C to 10°C in the Niyaqu basin (NAMORS 2018; Zhang et al., 2013).

## 3 Data

We use Sentinel-1 Level-1 single look complex data for all InSAR analysis, both from ascending and descending orbits from the interferometric wide swath mode with a ground resolution of 20 m azimuth and 5 m in range direction (ESA, 2012). We used a multi-looking factor of 4 in range direction and 1 in azimuth direction to achieve a ground resolution of 20 m. Sentinel-1 observes the earth's surface at an angle of 33 to 43° from the vertical (Yagüe-Martínez et al., 2016). SAR satellites are generally right-looking, meaning the microwaves are emitted to the right of the satellite. Due to the polar orbit, this causes the microwaves to be emitted in a near-east direction while the satellite is ascending and in a near-west direction during descending data acquisitions. The high elevation of the TP brings both advantages and disadvantages to InSAR applications. Large altitude variations can be problematic due to artifacts caused by atmospheric delay (Li et al., 2012), while the lack of high and dense vegetation reduces the risk of decorrelation, which would otherwise lead to poor phase stability, so-called coherence.

Sentinel-1a has been acquiring data since October 2014 and Sentinel-1b since September 2016. We started our time series analysis of Qugaqie basin in May and November 2015 for ascending and descending acquisitions, respectively, due to low coherence in earlier acquisitions. Early data over the Niyaqu basin produces better interferograms, here we start our time series analysis already in December 2014 for both ascending and descending acquisitions. The latest acquisitions included in the analysis are from November and December 2018. Sentinel-1b data is not available for this region, except for a 3 months period at the end of 2016 in descending orbit. More detailed information about the number of acquisitions and interferograms is shown in Table 1. The temporal baselines of

interferograms are 12 to 60 days for the Niyaqu basin and 12 to 96 days for the Qugaqie basin. We increased the temporal baseline for the Qugaqie basin to avoid a temporal data gap during the summer months of 2016 and 2017 caused by interferograms with low overall coherence. Decorrelation can occur where the surface displacement is greater than half the wavelength between two acquisitions (Crosetto et al., 2016). This applies to areas with LOS velocities >17.0 cm yr$^{-1}$ in the Niyaqu basin and 10.6 cm yr$^{-1}$ in the Qugaqie basin. All topographic analysis and processing, including the removal of the topographic phase from the InSAR data was conducted, using the 0.4 arc sec, equal to 12 m at the equator, resolution TanDEM-X DEM (©DLR, 2017). This new and truly global DEM has been acquired in 2010 to 2015 using single-pass X-Band SAR interferometry (Zink et al. 2014) and finally released by German Aerospace Agency in 2017. On the global scale the DEM features an absolute error at 90% confidence level of <2 m (Wessel et al. 2018). In steep terrain accuracy is ensured by multiple data takes in ascending and descending orbits with varying incidence angles to prevent radar shadows and overlay. In the Niyaqu basin, the number of acquisitions per pixel ranges from 5 to 8, with the majority representing average height estimates based on 6 acquisitions. Here, the mean 1 σ height error is 0.30 m. In the steeper Qugaqie basin the number of acquisitions ranges from 8 to 12, with the majority at 9 acquisitions. Here, the mean of the 1 σ height error is 0.35 m.

*Table 1:* *Sentinel-1 data used for the time series analysis of both study areas.*

| Area of interest | orbit | Acquisition period | Acquisitions / interferograms | Temporal baselines | Incidence angle |
|---|---|---|---|---|---|
| Niyaqu | ascending | 2014-12-31 to 2018-12-22 | 79 / 244 | 12-60 days | 40-42° |
| Niyaqu | descending | 2014-12-14 to 2018-11-11 | 72 / 227 | 12-60 days | 39-41° |
| Qugaqie | ascending | 2015-06-05 to 2018-12-22 | 74 / 278 | 12-72 days | 36-37° |
| Qugaqie | descending | 2015-11-15 to 2018-12-29 | 63 / 257 | 12-96 days | 43° |

## 4 Methods

### 4.1 ISBAS Processing

There are many different InSAR techniques capable of time series analysis to determine surface displacement over time. We chose a modified version of the Small BAseline Subset (SBAS) method (Berardino et al., 2002), which we performed with the ENVI SarScape software (©Sarmap SA, 2001-2019). The SBAS method generates interferograms between SAR acquisitions with a temporal baseline under a chosen threshold and stacks them to estimate displacement and velocity over a

longer time period. Interferograms are a spatial representation of the phase difference of two SAR acquisitions and can be used to determine the relative surface displacement between them. The phase stability, so-called coherence, is often used to represent the quality of an interferogram and to

215 determine which pixels will be processed further (Crosetto et al., 2016). The modified SBAS approach we employ, referred to as Intermittent SBAS (ISBAS), produces an improved spatial coverage by allowing limited interpolation of temporal gaps for areas, where the coherence is intermittently below the chosen threshold (Sowter et al., 2013; Batson et al., 2015). This reduces one of the downsides of the original SBAS algorithm, where partially vegetated areas can often not be

processed, due to the poor coherence induced by vegetation. We chose a coherence threshold of 0.3 for our velocity models with an intermittent value of 0.75 and therefore 75 % of the interferograms need to produce a coherence of at least 0.3 to be considered during unwrapping. These parameters are similar to those used by Sowter et al. (2013) and Bateson et al. (2015) and produce an acceptable compromise of good spatial coverage, while excluding most unreliable data from the unwrapping

process. We carefully analyzed all individual interferograms and excluded those with unwrapping errors and overall low coherence and therefore poor spatial coverage.

The topographic phase was removed from the interferograms with the TanDEM-X 0.4 arcsec resolution DEM (Wessel et al., 2018) and the orbital phase was removed by subtracting a constant simulated phase from our interferograms. We then estimated and subsequently subtracted a $3^{rd}$

order polynomial function over flat stable areas to remove any remaining large scale phase ramps. To reduce spatial trends connected to the small size of the Qugaqie basin we processed a larger area which also includes the two neighboring catchments during the ISBAS workflow. Zhao et al. (2016) demonstrated, that using a linear model to process regions with cyclical heave-subsidence mechanisms leads to an overestimation of the displacement signal. We could not confirm their

findings in our study areas. We therefore decided to use a linear model for all processing, as the quadratic model produced almost identical results and the cubic model produced unreliable results with poor spatial coverage. We applied a short atmospheric high pass filter of only 100 days, to preserve the seasonal signal for our time series analysis, and a low pass filter of 1200 m.

After performing the ISBAS processing chain, flat areas within Qugaqie basin retained a relatively

strong spatial trend of up to 9 mm $yr^{-1}$ and 13 mm $yr^{-1}$ in ascending and descending datasets, respectively. This signal is likely connected to an atmospheric phase delay rather than actual surface displacement. We therefore performed a linear trend correction to remove this spatial trend from both ascending and descending datasets. The linear spatial trend was estimated through likely unmoving areas with very low slope of <5° with at least 200 m distance to water bodies (based on

NDWI of Sentinel-2 optical imagery, ©Copernicus Sentinel data 2018, processed by ESA). After these

corrections we performed a decomposition of ascending and descending data sets where we assume displacement in the north-south direction to be insignificant, to determine vertical and east-west displacements. We observe insignificant mean east-west velocities of -0.2 mm yr$^{-1}$ and -0.9 mm yr$^{-1}$ with standard deviations of 2.2 mm yr$^{-1}$ and 2.4 mm yr$^{-1}$ in likely stable areas in Niyaqu and Qugaqie basin respectively.

## 4.2 Selection of reference areas

InSAR displacement products are spatially relative to the chosen reference areas. It is necessary to select at least one reference to perform the unwrapping process during the InSAR processing chain. Stable GNSS stations are preferred reference points but there are no permanent GNSS stations installed near the study areas. Therefore it is necessary to select the stable reference areas carefully to avoid introducing an erroneous trend signal into the surface displacement models. The parameters by which those stable reference areas were chosen are:

1.) The reference points must be at locations which are represented clearly in 100 % of all interferograms generated during the SBAS processing chain, to ensure that the displacement of all interferograms can be correctly determined relative to those points.

2.) Whenever possible we selected bedrock at high elevations far away from the valley floor. The annual heave-subsidence cycle, and the corresponding uplift and subsidence of the ground, is very strongly represented in the highly moisturized ground of the valley floor. Choosing reference points in this environment would remove this annual ground oscillation from the dataset in the valley floor and create an artificial and opposite oscillation pattern in other areas. Bedrock has a much smaller porosity than loose sediment or soil and is therefore less prone to oscillations forced by freezing and thawing of pore fluid. Stable bedrock is associated with a high coherence throughout the year, due to its relatively stable backscatter characteristics

3.) The chosen reference points must be stable during the entire period of observation, as moving reference points would shift the entire velocity model. We compared the results of different reference points in areas where we expect little motion and discarded those that caused a shift. As reference areas we chose regions with a low slope, good coherence and no obvious deformation structures, and assume them to be stable in time.

Despite our careful selection of reference points, we cannot be certain, that those areas are in fact stable throughout the entire data acquisition period. We therefore chose to use multiple reference points instead of a single point to produce the surface velocity models. This prevents a single, potentially poorly selected, reference point from invalidating the entire dataset by introducing either

a multiannual velocity shift or seasonal displacement signal. The areas of partially exposed bedrock and the mountainous terrain of the Qugaqie basin made the selection of reference points easier compared to the Niyaqu basin, where exposed bedrock is rare. Selecting only points positioned at these optimal locations left us with none near the center of the basins or the lake shore. This caused velocity shifts along the LOS on a millimeter scale in presumably stable flat areas, if they were far away from the reference points. We therefore increased the number of reference points to 90 and 51 in the Qugaqie basin and 92 and 61 in the Niyaqu basin for ascending and descending acquisitions respectively. Maps showing their locations are included in the supplement. The number of reference areas varies between ascending and descending acquisitions, due to differences in coherence, but we chose the same reference areas whenever possible.

## 4.3 Surface displacement models

In total we produced three different types of surface displacement models for the Niyaqu and the Qugaqie basin: The Linear Velocity Model (LVM), the Heave-Subsidence Model (HSM) and the Seasonal Slope process Model (SSM). The LVM portrays the mean surface velocity from 2015 to 2018 and does not portray seasonal variations. It describes both valleys and slopes, but we make the assumption that displacements are predominantly vertical in areas with slopes <10° and orientated in a downslope direction in steeper areas. The HSM models the heave-subsidence cycle caused by freezing of the active layer in autumn followed by subsequent thawing of the active layer in spring. It covers only areas with slopes <10°, shows only seasonal displacement and assumes that all displacement is vertical. The SSM focuses on slopes where sliding is accelerated from spring to autumn, to be differentiated from slopes in the LVM where sliding takes place throughout the year at a near linear rate. It only covers slopes >10° and assumes that all displacement is orientated in a downslope direction.

*Table 2:* *Overview of the 3 surface displacement models with information regarding their purpose, displacement patterns and their connections to geomorphological und geological parameters.*

| Model type | LVM (Linear Velocity Model) | HSM (Heave-Subsidence Model) | SSM (Seasonal Slope process Model) |
|---|---|---|---|
| **Purpose** | Multiannual subsidence, sediment accumulation and permafrost creep | Seasonal heave-subsidence cycle due to freezing and thawing of the active layer | Seasonally accelerating slope processes |
| **Displacement type** | Multiannual linear velocity | Seasonal vertical | Seasonal displacement |

| | along the slope or vertical | displacement | along the slope |
|---|---|---|---|
| **Slope** | <10°: vertical velocity <br> >10°: along slope velocity | <10° | >10° |
| **Material** | Soil, regolith, till, debris and ice | Mainly soil | Regolith, debris and ice |
| **Related geomorphological processes** | Long-term subsidence, sediment accumulation and permafrost creep | Heave-subsidence cycles connected to cryoturbation | Solifluction, gelifluction and rock slope instability on seasonally frozen slopes |
| **Associated Landform** | Rock glaciers, protalus ramparts and moraines | Valley bottom terrain | Debris mantle slopes, solifluction lobes and rockslides |

### 4.3.1 Linear Velocity Model (LVM)

This model portrays the mean annual surface velocity, with different methods applied to regions with a slope >10° and with a slope <10°. Seasonal displacements trends are not present in this model, as we address those in the separate models HSM and SSM. The original ISBAS processing chain (Sect. 4.1) is the same but we applied different methods to project the LOS results into a more meaningful direction. For areas with a slope <10°, we assumed that displacement would occur mainly in a vertical direction, as the slope would be too small to facilitate significant sliding or creep in most cases. Matsuoka (2001) shows that while solifluction has been documented on slopes as low as 2°, most affected areas in mid-latitude to tropical mountains (including the TP) feature slopes >10°. To determine the vertical velocity, we performed a decomposition of ascending and descending time series data. For this process we assume the north-south component of the surface displacement to be zero, which allows us to determine the vertical and east-west components (Eriksen et al., 2017). The vertical component represents our expected surface velocity for flat areas, while the east-west component can be used to assess the error range of the velocity model.

The decomposition method works well for flat regions and slopes with an east or west aspect but does not produce useful data for slopes with a north or south aspect. The Sentinel-1 SAR satellite constellation is quite sensitive to both east-west and vertical surface displacement, but very insensitive to displacement with a strong north or south component. This is problematic when studying displacements with a large horizontal component, as the velocity of surfaces moving in a

northern or southern direction will be either severely underestimated or completely overlooked. We therefore employed a different method for slopes. Areas with a slope >10° were projected in the direction of the steepest slope, as most surface displacement is assumed to be caused by sliding processes transporting material parallel to the slope. We made an exception for areas with an east-west velocity >10 mm yr$^{-1}$, as our study areas feature a periglacial setting with landforms such as rock glaciers, which move in a downslope direction and may extend into flatter areas. Those areas were projected in a downslope direction, even on slopes <10°. This approach (Notti et al., 2014) originated from landslide studies to produce a more accurate result for a process, where the direction of the moving structure is either known or can be assumed with reasonable certainty.

To estimate the downslope velocity, we calculate a downslope coefficient, with values between 0.2 and 1 and divide the LOS velocity by this coefficient to determine the downslope velocity. Maps of the spatial distribution of this coefficient are included in the supplement. We used a smoothed version (90 x 90 m moving mean) of the TanDEM-X DEM to determine the motion direction, as we assume that structures such as rock glaciers and landslides move a larger amount of sediment in a similar direction. Small scale variations of the aspect or slope have a strong impact on the downslope coefficient and would create outliers in the slope projection in areas with high surface roughness. It is important to note, that by projecting LOS velocities along the steepest slope, we not only assume the direction vector, but we also simplify the mechanics to that of a planar slide. In doing so we assume that neither rotational nor compressing processes are involved. This is an obviously unrealistic but necessary simplification, which leads to on overestimation of the downslope velocity. The error range of the slope projection can be up to 5 times higher for areas with a very strong downslope coefficient than the range of ±2.4 mm we determined over flat ground.

**4.3.2 Heave-Subsidence Model (HSM)**

Prior to analyzing the heave-subsidence amplitude, we projected the LOS displacements from both ascending and descending datasets to vertical displacements. We then removed the linear multiannual trend from the datasets to isolate the seasonal signal. Sum of sine functions were estimated for each individual time series. For the amplitude estimation in the Qugaqie basin we used a function with two sine terms and for Niyaqu basin a function with three terms. We identified the sine term representing the seasonal signal and discarded the other terms. We calculated the mean values and the standard deviation of the amplitude, shift and period of the sine curve during the 3 to 4 seasonal cycles. We use the standard deviation of the amplitude and the shift as a measure of their error ranges. We also calculated the explained variation ($R^2$), which represents the proportion of variation of the time series explained by the sine curve regression. To qualify for further analysis, a time series must display a heave-subsidence amplitude larger than 3 mm, with an $R^2$ >0.5 and a

period of 350 to 380 days. The main results of the HSM are the mean amplitude of the heave-subsidence cycle and the Day of Maximum Subsidence (DMS). The DMS describes the mean day on which the sine function reaches its minimum. This represents the day on which the soil has subsided to its minimum level due to thawing before beginning to heave again due to freezing. The final HSM contains both ascending and descending data. In areas where they overlap we show the mean value of the two. Slopes >10° were excluded from the HSM as these areas are likely to display mainly gravity-driven displacement with a downslope direction and not only vertical heave-subsidence cycles. The seasonal displacement of slopes is covered by the SSM and their multiannual velocity is shown in the LSM instead.

### 4.3.3 Seasonal Slope process Model (SSM)

The average seasonal velocities represent the median summer and median winter velocities over the entire time series. We divided the median summer velocity by the median winter velocity to produce the seasonal sliding coefficient, which represents how fast a surface is moving in summer compared to winter. Our SSM features a precision (1 sigma) of around 2.4 mm yr$^{-1}$. Time series with median seasonal velocities <2 mm yr$^{-1}$ were set to 2 mm yr$^{-1}$, to avoid artificially large values when calculating the seasonal sliding coefficient with median seasonal velocities close to 0. This affects 18.1 % of slopes in Niyaqu basin and 4.3 % of slopes in Qugaqie basin. The higher value in the Niyaqu basin is due to the lower overall velocity of slopes in that region and the reduced spatial coverage due to lower coherence in the higher zone where larger velocities occur. A seasonal sliding coefficient of 1.5 represents a 50 % increased summer velocity compared to the winter velocity. We chose this threshold of 1.5 to differentiate between seasonally accelerated slopes and slopes with relatively linear velocity.

## 5 Results

### 5.1 Linear surface velocity derived from LVM

Solifluction may occur on slopes as low as 2° but mostly affects areas with a slope >10° in mid-latitude to tropical mountain areas (Matsuoka 2001). This is corroborated by the east-west velocity produced by our decomposition of ascending and descending data. For the slopes of 0 - 5°, 5 - 10° and 10 - 15° we observe mean east-west velocities of -0.1 mm yr$^{-1}$, -0.6 mm yr$^{-1}$ and -0.6 mm yr$^{-1}$ at standard deviations of 3.0 mm yr$^{-1}$, 2.9 mm yr$^{-1}$ and 5.0 mm yr$^{-1}$ respectively. The jump in standard deviation from 5 - 10° to 10 - 15° from 2.9 mm yr$^{-1}$ to 5.0 mm yr$^{-1}$ suggests that we observe considerably more horizontal displacement in the latter group. This makes the 10° mark a good

threshold between the vertical and the downslope projections. For areas with a slope >10° we assumed that the displacement would occur along the steepest slope, driven by gravitational pull (Haeberli et al., 2006). Unconsolidated material and the lack of deep-rooted vegetation in the area (Li et al., 2014) facilitate downslope motion.

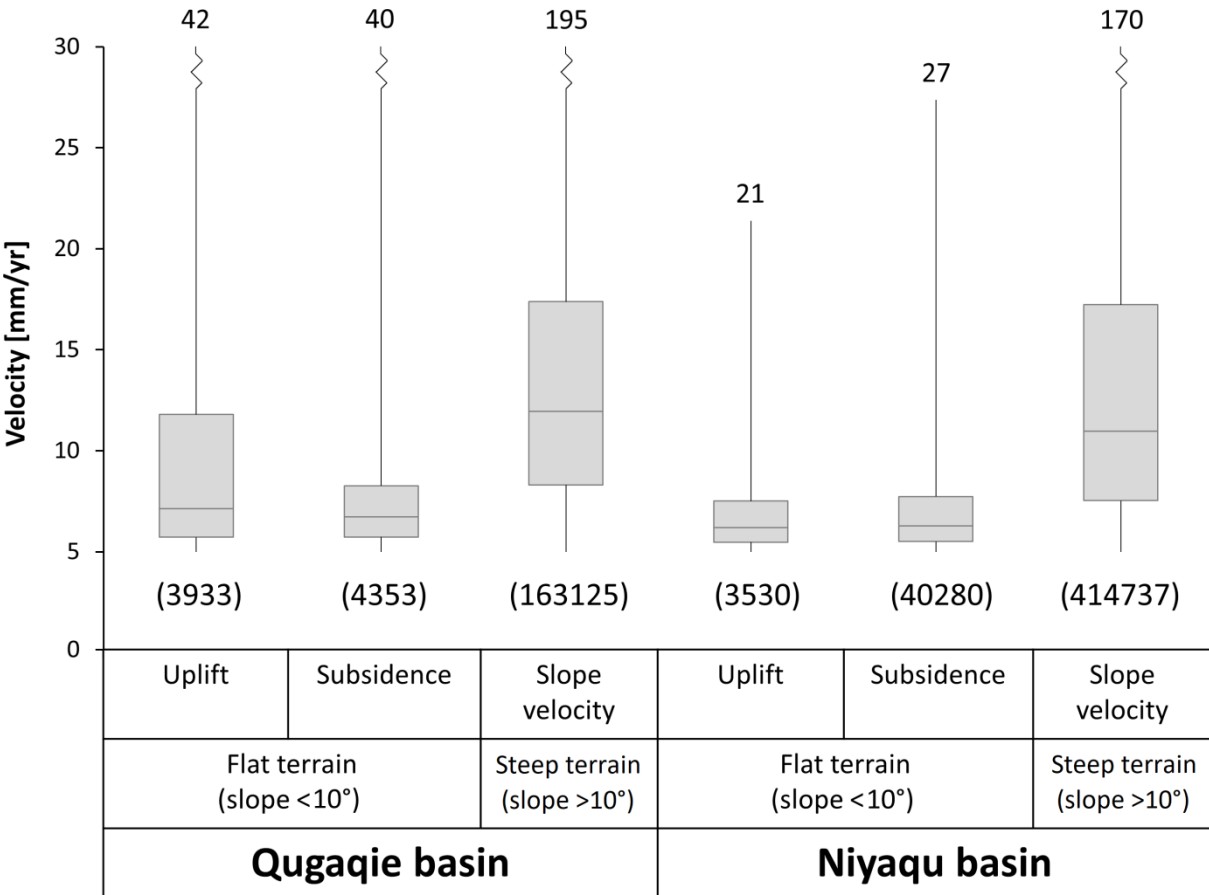

*Fig. 2: Distribution of the mean velocity results of the LVM for unstable flat and steep terrain in both study areas. All surface motion in flat areas (slope <10°) was projected into vertical direction (uplift and subsidence) and motion in steeper areas was projected along the direction of the slope. The maximum values are shown above the respective boxplots and the amount of data points included in each plot is shown in parenthesis below them. Areas with velocities <5 mm yr$^{-1}$ are considered stable* 400 *and are not shown here.*

Spatial data gaps in our InSAR models are caused by layover and shadow effects in mountainous regions or where the coherence was lost due to streams, vegetation, rock falls and glaciers. These data gaps make up 34.7 % in flat and 31.4 % in steep terrain within the Qugaqie basin and 30.5 % and 36.0 % in the Niyaqu basin. The decomposition of ascending and descending datasets of areas with 405 flat terrain (slope <10°) shows that both basins have relatively stable flat terrain on a multiannual scale. 53.3 % of flat areas in the Qugaqie basin and 64.4 % in Niyaqu basin fall within the ±5 mm yr$^{-1}$ velocity group in both vertical and east-west directions. We consider these areas to be stable. In the

Quagqie basin, 3.3 % of flat areas experience uplift, most of which are near the main stream, while 2.8 % of flat areas are subsiding. In the Niyaqu basin, 0.2 % of flat areas experience uplift and 2.7 % experience subsidence. The remaining flat areas, 5.8 % in the Quagaqie and 2.1 % in the Niyaqui basin, experience minor horizontal motion.

*Table 3: Summary of the spatial data coverage of the LVM in the study areas. The values represent the percentages compared to all flat or steep terrain in the respective study area. Incoherent areas display a mean coherence of <0.3. Stable areas are characterized by multiannual velocities <5 mm yr⁻¹ in all directions. Unstable flat areas move at >5 mm yr⁻¹ and are divided into uplift/ subsidence/ horizontal motion. Steep unstable areas move at >5 mm yr⁻¹ downslope and very unstable terrain moves at >30 mm yr⁻¹.*

|  | Terrain | Incoherent | Stable | Unstable | Very unstable |
|---|---|---|---|---|---|
| **Qugaqie** | flat (<10°) | 34.7 | 53.3 | 3.3/2.8/5.8 | 0.1 |
| **basin** | steep (>10°) | 31.4 | 20.9 | 44.9 | 2.8 |
| **Niyaqu** | flat (<10°) | 30.5 | 64.4 | 0.2/2.7/2.1 | 0.0 |
| **basin** | steep (>10°) | 36.0 | 21.1 | 39.7 | 3.1 |

Steep terrain is considerably more unstable in both study areas. In the Qugaqie basin only 20.9 % of areas in steep terrain are stable, with 2.8 % being very unstable with velocities >30 mm yr⁻¹. In the Niyaqu basin 21.1 % of areas in steep terrain are stable and 3.1 % are very unstable. A summary of the spatial data coverage is shown in Table 3. A distribution of the absolute surface velocity results in different regions is shown in Figure 3.

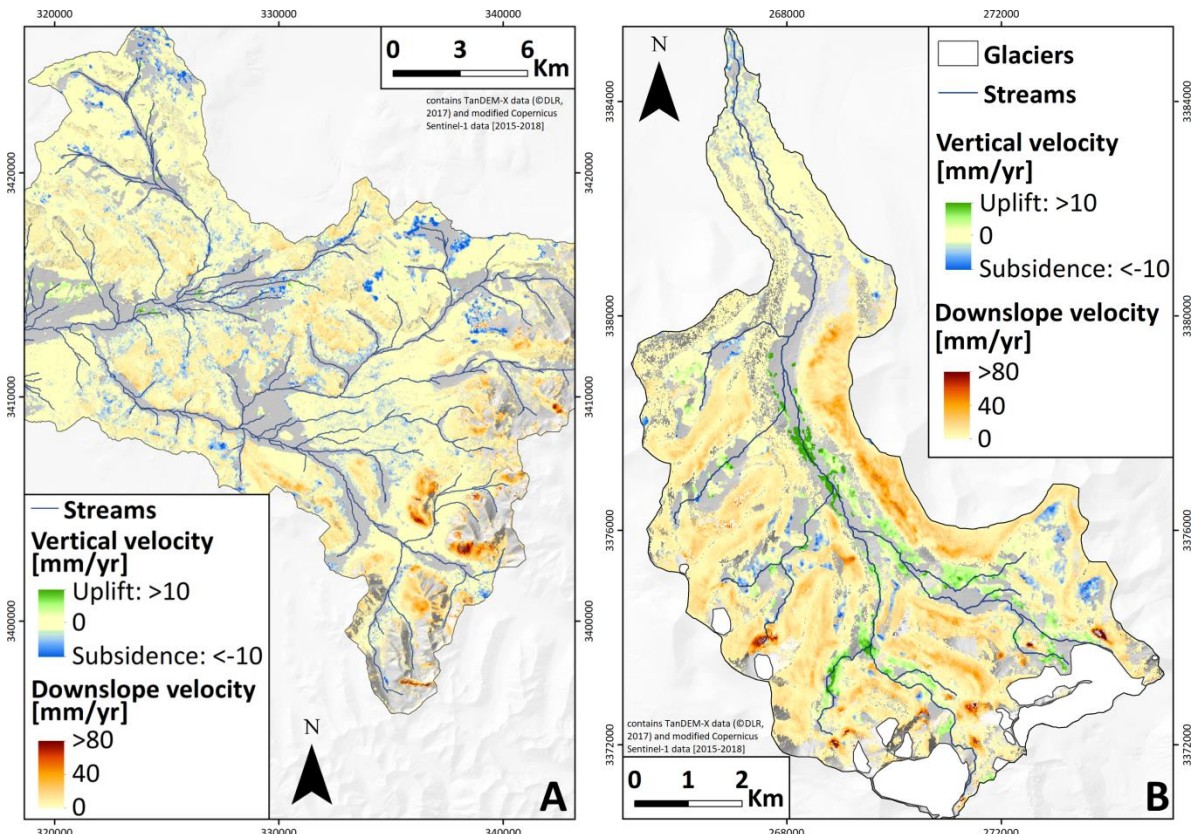

**Fig. 3**: *LVM of the Niyaqu **(A)** and Qugaqie **(B)** basins based on Sentinel-1 data (©Copernicus 2015-2018) over TanDEM-X DEM (©DLR, 2017). Flat terrain with a slope <10° show vertical velocity and steeper terrain show the surface velocity projected along the steepest slope. Maps showing the spatial distribution of flat and steep terrain are included in the supplement.*

The coherence in both basins is much reduced in valley bottoms. Streams and other water bodies affecting as well the soil moisture status of the neighboring land surfaces cause large changes in microwave backscatter properties depending on the season. More extensive vegetation near the valley bottom further reduces the coherence. The coherence is especially low in valley bottoms during the spring and the summer monsoon period, when the ground thaws, the surface is inundated by rain water or runoff and biomass production increases. This causes an overall drop in spatial data coverage in valley bottoms, as many resolution cells exhibit coherence values below the threshold. Coherence maps of ascending and descending orbit of both study sites are included in the supplement.

### 5.2 Heave-subsidence cycle derived from HSM

The seasonal vertical oscillation of the ground due to freezing and thawing of the soil is strongest in the valley bottom, especially near streams, lakes, ponds and, in the case of Qugaqie basin, glaciers (Fig. 5 A/B). In these areas the amplitude of this oscillation can reach up to 19 mm in the Qugaqie basin or even 27 mm in the Niyaqu basin. The median amplitude error is 1.3 mm in Niyaqu basin and

1.1 mm in Qugaqie basin. The Day of Maximum Subsidence (DMS) is the day in summer during which the soil has subsided to its minimum level before beginning to heave again in autumn (Fig. 5 C/D). In the Qugaqie basin the median DMS is on July 19 and in the Niyaqu basin it is on August 23 (Fig. 4). Most data points with heave-subsidence amplitudes of <7 mm reach their DMS in July to August in the Niyaqu basin and May to July in the Qugaqie basin, while areas with larger amplitudes tend to reach theirs in September to October (Fig. 5E/F). The median shift error of the sine function modeling the heave-subsidence cycle is 33 days in the Niyaqu basin and 27 days in the Qugaqie basin. We compared the DMS results of ascending and descending datasets and noticed that in both basins the mean DMS of the descending dataset occurs earlier. In the Niyaqu basin the difference between ascending and descending DMS is 27 days and in the Qugaqie basin 10 days. Maps showing the spatial distribution of this disparity are included in the supplement.

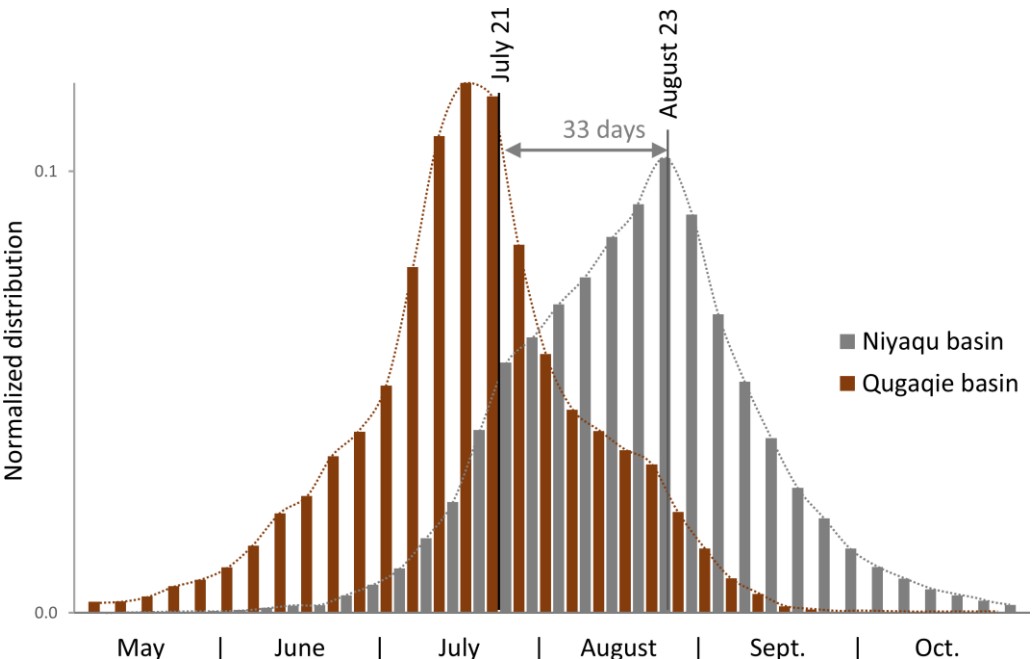

*Fig. 4:* Normalized distribution of the thaw-induced Day of Maximum Subsidence (DMS) in the Niyaqu basin (grey) with a median value on August 24 and the Qugaqie basin (brown) with a median value on July 21. The lag time of 33 days between the median value of Niyaqu basin and the day of the mean maximum air temperature on July 21 (NAMORS, 2018) is also shown. The median DMS of Qugaqie basin occurs on July 19 resulting in no clear lag time.

The freezing and thawing of soil follows the Stefan equation (Riseborough et al., 2008) and there is a significant lag time between the day of maximum air temperature and the DMS. This lag time has been studied with InSAR remote sensing techniques both on the northern and the southern TP (Li et al., 2015; Daout et al., 2017). According to the weather data from the NAMORS research station, the

air temperature has a mean peak on July 21 from 2010 to 2017. Data from the weather station at the Zhadang glacier (Zhang et al., 2013) shows this mean peak on July 27 for 2010 to 2011 (July 19 for the same period at NAMORS). NAMORS research station is close to Niyaqu basin but about 50 km distant from Qugaqie basin and at lower altitude. The Zhadang glacier and its weather station are located within Qugaqie basin. Due to the very short acquisition period of only two years for the Zhadang

weather station, we chose the eight year data set of the NAMORS for both study areas. This produces a lag time of approximately 33 days for Niyaqu basin (Fig. 4, grey), while in the Qugaqie basin the median DMS occurs on July 19, two days before the maximum air temperature on July 21, resulting in no clear lag time (Fig. 4, brown).

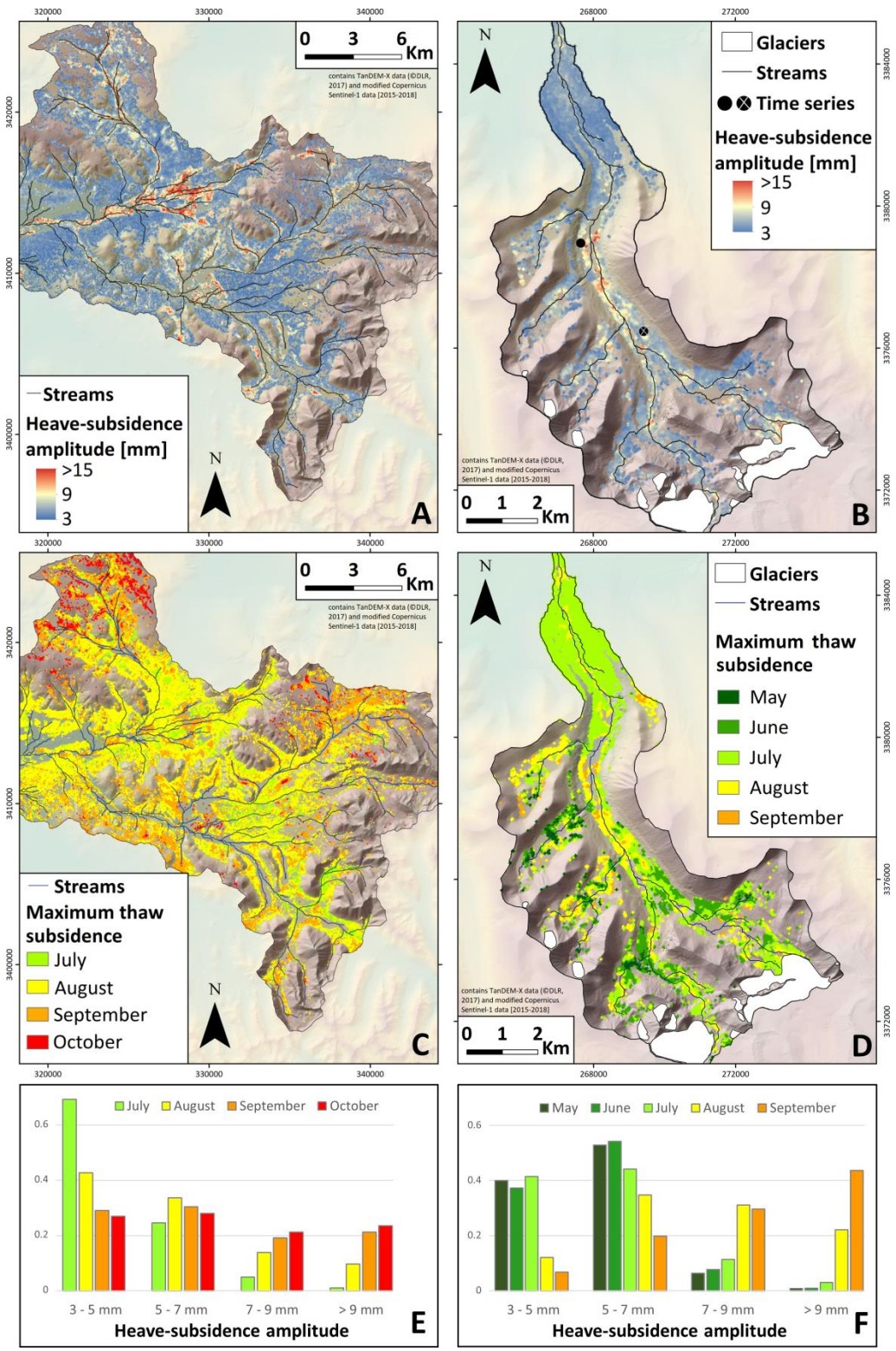

***Fig. 5****: Parameters of the HSM (modified Copernicus Sentinel-1 data [2015-2018]) over TanDEM-X DEM (©DLR, 2017). Spatial variations of the mean amplitude (**A/B**) and the day of maximum subsidence (**C/D**) of the HSM within the Niyaqu/Qugaqie basin. The locations of the time series of Figure 6A are displayed as black dots.  **E/F:** Normalized distribution of the months in which the heave-*

*subsidence cycle reaches their maximum subsidence split up into 4 groups according to their amplitude for the Niyaqu/Qugaqie basin.*

## 5.3 Seasonally accelerating slopes derived from SSM

We identified two distinct seasons, the wet monsoon season in summer and the dry winter season, which have a clear impact on the displacement data. The former season causes accelerated ground sliding on many slopes, while the latter slows most sliding processes (Fig. 6C). We refer to this

seasonally accelerated sliding as freeze-thaw-driven but it is possible that the water input during the monsoon period, which coincides with the highest annual air temperatures, amplifies this process. In the Niyaqu basin the accelerated displacement pattern of the summer period lasts from May to September and in the Qugaqie basin from June to October. The slower winter displacement patterns last from November to March and from December to April, respectively. We compared the median

summer velocities to the median winter velocities of each time series over the entire study period. Most soil or debris covered slopes in both basins display accelerated sliding rates of 100 to 300 % towards the end of the summer monsoon. They reach downslope velocities of mostly 50 to 150 mm $yr^{-1}$ during that time.  The lower areas of most slopes appear to move at a linear rate. This marks the interface between the heave-subsidence cycle of the valley bottoms and the seasonally accelerated

sliding on the slopes. These two seasonal displacement processes (Fig. 6A/C) are both present in those interface areas and often interfere with each other in such a manner, that they appear to move linearly in the SSM.

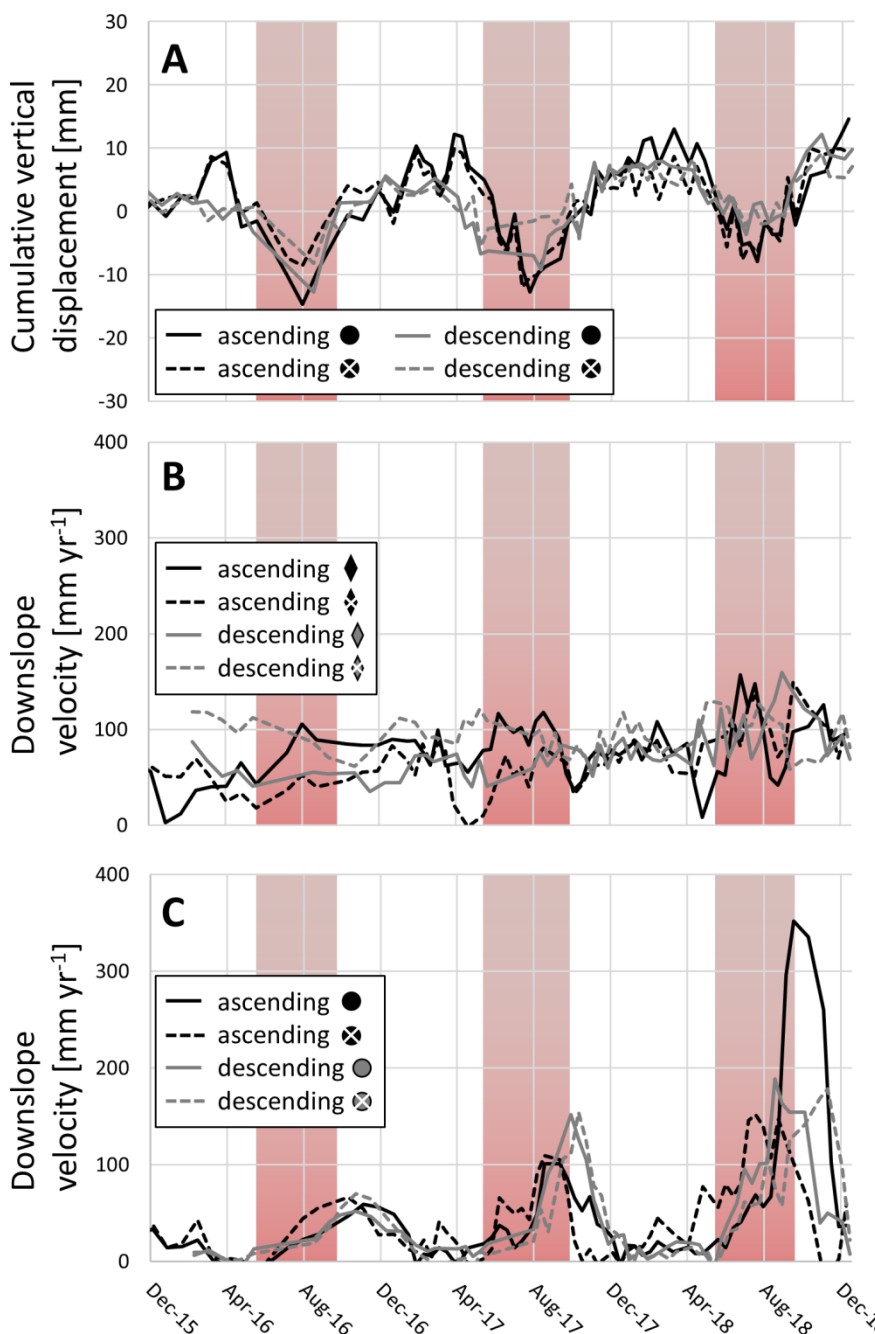

***Fig. 6****: Surface displacement time series of both ascending (black, dashed black) and descending (grey, dashed grey) data of various areas throughout Qugaqie basin highlighting the three seasonal patterns. June to September are shown with a red background as they display the strongest monsoon activity and air temperatures >0 °C (Zhang et al., 2013).* ***A:*** *Cumulative vertical displacement showing the seasonal heave-subsidence cycle of the HSM at two locations near the stream of the main valley (black dots in Fig. 5B).* ***B:*** *Downslope velocity time series of the four gravity-driven landforms (rhombi in Fig. 7A) with relatively constant velocities (blue areas in Fig. 7A).* ***C:*** *Freeze-thaw-driven displacement patterns on four slopes (dots in Fig. 7A) with accelerated displacements in summer and comparatively minor displacements in winter (red areas in Fig. 7A). B and C display moving average values of the closest 4 values in time.*

In the Qugaqie basin and to a lesser extent in the Niyaqu basin, we observe that some of the fastest
moving structures creep at a linear rate, as opposed to the strong seasonality of most slopes. They
do not display a clear acceleration in summer and their multiannual velocity is generally between 30
to 180 mm yr$^{-1}$ (Fig. 6B). We identified some of these landforms as rock glaciers or protalus ramparts
from optical satellite imagery, field observations and topographic analysis. The motion of these
permafrost related landforms, is driven by an ice matrix in between unconsolidated debris material
(Haeberli et al., 2006). Rock glaciers and protalus ramparts represent 36 % of the linearly fast moving
landforms in the Qugaqie basin and include the largest and the fastest landforms with this
displacement pattern. Other landforms with this displacement pattern are rock slope instabilities (24
%) and frozen moraines (39 %). Example pictures of those three landform types are included in the
supplement. We focus our analysis of slope displacements on the Qugaqie basin, due to the
considerably better overall coherence and therefore spatial coverage of slopes in the periglacial
zone. The maps equivalent to Figure 7 for the Niyaqu basin characterized by poor coherence in the
periglacial zone can be found in the supplementary material.

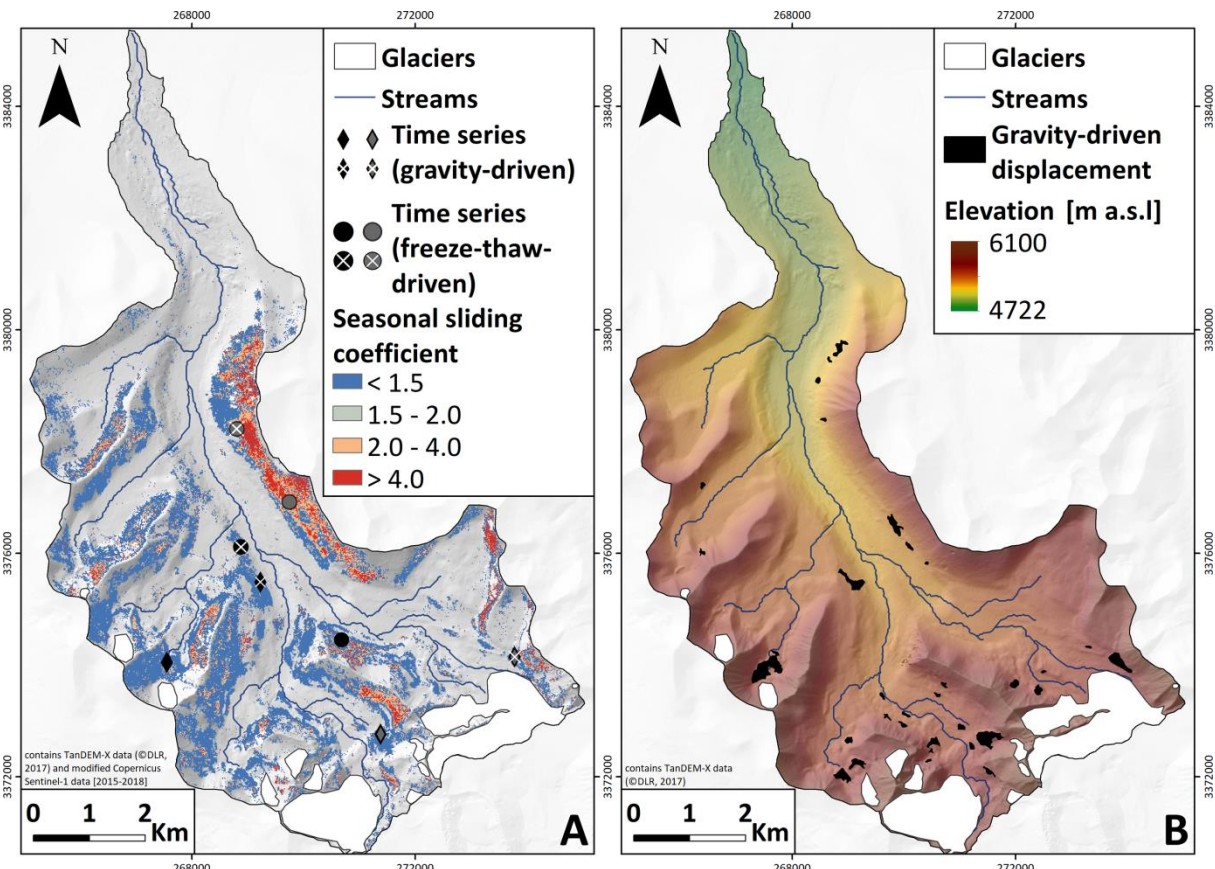

***Fig. 7: A:*** *SSM of Qugaqie basin displaying the spatial distribution of slopes with accelerated surface
velocity in summer compared to winter. A seasonal sliding coefficient of 1.5 represents a 50 %
increase of the velocity in summer. Only areas with a slope >10° and a slope velocity >10 mm yr$^{-1}$ are
shown. The locations of the time series in Figure 6B/C are shown as rhombi/dots.* ***B:*** *Spatial*

*distribution of clusters where we assume displacement to be gravity-driven. These clusters display slope velocities >50 mm yr$^{-1}$ and a seasonal sliding coefficient <1.5.*

## 6 Discussion

We created three different models to study both seasonal and multiannual surface displacements and their driving processes in steep and flat terrain at Nam Co. We discuss the multiannual displacements of the LVM and the seasonal displacements of the HSM in flat terrain in Sections 6.1 and 6.2 respectively. Multiannual displacements of the LVM and seasonal displacements of the SSM in steep terrain are discussed in Sections 6.3 and 6.4 respectively.

For both our seasonal and our multiannual surface displacement models we distinguish between flat terrain with slopes <10° where we assume all displacement to be mainly vertical and steep terrain with slopes >10° where we assume displacement to occur in a downslope direction. We chose this threshold based on the decomposition of ascending and descending results, which shows a considerable increase of horizontal velocities on slopes >10° compared to flatter areas. While this simplification is necessary and mostly accurate for our study areas, it leads to inaccuracies especially in areas with slopes of 5 to 15° where displacements can occur in both vertical and downslope directions.

### 6.1 Multiannual displacements in flat terrain

Flat terrain throughout both basins can be considered relatively stable with mean vertical and east-west velocities within ±5 mm yr$^{-1}$. Some areas in valley bottoms show uplift rates of up to 40 mm yr$^{-1}$ (Qugaqie basin, Fig. 3B) and 21 mm yr$^{-1}$ (Niyaqu basin, Fig. 3A). A possible explanation is that seasonal variations in the soil moisture content were misidentified as surface displacement. The melt water and high monsoon precipitation increases the soil moisture in summer, which can lead to a false interpretation of surface displacement of 10 to 20 % (Zwieback et al., 2017). In our case this would represent 5 to 10 mm per monsoon season for a total of 15 to 30 mm. This is corroborated by the large amplitude of the heave-subsidence model in the same areas, which is associated with more water during the freezing process (Fort and van Vliet-Lanoe, 2007).

We also observe subsidence rates of up to -12 mm yr$^{-1}$ in Qugaqie basin and -25 mm yr$^{-1}$ in Niyaqu basin. Many of those pixels are close to streams and water bodies. Approximately 30 % in the Niyaqu and 60 % in the Qugaqie basin fall into permafrost regions (Zou et al., 2017; Tian et al., 2009). This makes permafrost degradation a potential driver of this subsidence, as a thinning permafrost layer would result in melt water escaping from the thawing soil. However, longer periods of observation are needed to come up with reliable conclusions.

## 6.2 Seasonal displacements in flat terrain

Both the Niyaqu and the Qugaqie basin have relatively stable flat terrain on a multiannual scale but show a strong seasonal signal in the same areas (Fig. 6A). It is unlikely that this signal is induced by seasonal atmospheric effects, as the amplitude would likely correlate to some degree with relative elevation to the reference points (Dong et al., 2019), which is not the case for our data. The most likely explanation is, that this signal represents the heave-subsidence cycle of freezing and thawing of moist soil. Others observed very similar signals over permafrost areas on the northern TP (Daout et al., 2017) and over seasonally frozen ground in Dangxiong county on the southern side of the Nyainqêntanglha range (Li et al., 2015). The heave-subsidence amplitude of these studies agrees well with our results of mostly 3 to 15 mm in both basins. In similar areas they observe amplitudes of 2.5 to 12 mm and 10 to 25 mm respectively. Comparable studies of permafrost landscapes measured thaw subsidence of predominantly 10 to 70 mm and 2 to 68 mm per thawing season on Spitsbergen Island (Rouyet et al., 2019) and in northern Siberia (Antonova et al., 2018), respectively.

In both of our study sites we observe a relation between the amplitude of the heave-subsidence cycle and the DMS (Fig. 5E/F). Areas with high amplitudes tend to reach the DMS later in the year (September to October) and areas with small amplitudes tend to reach their DMS earlier (July to August). The lag time between maximum air temperature and DMS is therefore greater for areas with a large heave-subsidence amplitude. Longer lag times have been associated with a deeper active layer when assuming a one-dimensional heat transfer in soils (Li et al., 2015). For our study areas this would imply, that the active layer tends to be deeper near streams and water bodies, as both the heave-subsidence amplitude and the lag time are highest there. This agrees with other studies on active layer thickness, which also observe a deeper active layer in those areas, especially where water bodies remain partially unfrozen in winter (e.g. McKenzie and Voss, 2013).

Other studies observed considerably higher lag times between the highest air temperature and the DMS. They observe lag times of 97 days and 65 days in flat areas (Daout et al., 2017; Li et al., 2015) with longer lag times in mountainous areas, which they attributed to thicker permafrost, colder surroundings and less soil moisture. A small amount of the data points of Li et al. (2015) fall within the Qugaqie basin, showing lag times of 50 to 90 days. In Qugaqie basin we observe no clear median lag time (Fig. 4). It is possible, that the difference between their results and ours is reflecting actual changes to the lag time between their dataset of 2007 to 2011 and ours of 2015 to 2018 but the point density of their data within Qugaqie basin is too low to draw reliable conclusions. We determined the lag time by comparing the DMS to the average maximum air temperature at NAMORS from 2010 to 2017. NAMORS is located at an elevation of 4730 m, which is considerably lower than most valley areas of Qugaqie basin at up to 5600 m. The temperature data acquired by

Zhang et al. (2013) within the Qugaqie basin shows a maximum air temperature on July 27, 6 days later than at NAMORS. Their data set covers a period of less than two years and is therefore likely too short for an accurate comparison but it indicates that the maximum air temperature is similar to that of NAMORS. The Qugaqie basin has a short thaw period, with only 80 to 100 days in summer reaching daily air temperatures >0° (Zhang et al., 2013). Together with the presence of permafrost this may explain the short lag time between the maximum air temperature and the DMS. A short thaw period is associated with a thinner active layer (Åkerman and Johansson, 2008) and the cold mountain climate and the permafrost would accelerate the freezing process of the active layer. While this may help to explain the short lag time, it also highlights a major limitation of the model. For the HSM we estimated a sine function for every individual time series to determine the spatial distribution of the heave-subsidence cycle. Areas where this cycle does not follow a sinusoidal pattern are therefore not represented accurately by the HSM. The short thaw period of the Qugaqie basin also shortens the periods of maximum subsidence compared to the periods of maximum heave (Fig. 6A), hence making the heave-subsidence cycle less sinusoidal. Together with the short time series of only 3 to 4 years, this leads to a high shift error of the sine function of 27 days in the Qugaqie basin and 33 days in the Niyaqu basin.

The difference of the DMS between ascending and descending data sets is 10 days in the Qugaqie basin and 27 days in the Niyaqu basin and not distributed randomly. This is a very high disparity for a displacement with a predominantly vertical direction, which should be represented equally in ascending and descending data sets if the incidence angles are comparable. The difference in incidence angles between the two orbits is 6° over the Qugaqie basin and 1° over the Niyaqu basin. Over the Qugaqie basin the resulting difference in sensitivity to vertical displacement and the different number of summer seasons during the observation period (4 for ascending and 3 for descending) may help to explain the 10 day disparity between ascending and descending data sets. This does not explain the large disparity of 27 days in the Niyaqu basin, as the difference in incidence angle is very small and same time period is covered by ascending and descending data sets. For Niyaqu basin in particular it was necessary to select different reference areas during interferogram generation, as the same areas often did not feature a high coherence in both orbits. Minor displacement signals of reference areas which were used in only one of the orbits may explain the widespread disparity of the DMS in the Niyaqu basin. The imperfect manner in which the sine curve of the HSM estimates the heave and subsidence of the ground also introduces uncertainties, as this disparity drops to 21 days in areas of Niyaqu basin where the explained variation $R^2$ >0.9 and rises to 29 where $R^2$ <0.6. The high disparity between the DMS results of ascending and descending data sets and the high shift error of the sine function suggest that the sinusoidal HSM does not produce reliable results of the DMS for such a short time series with only 3 to 4 seasons.

## 6.3 Multiannual displacements in steep terrain

Most data points on slopes in both basins show downslope velocities of 8 to 17 mm yr$^{-1}$ with a small number of landforms moving faster than 30 mm yr$^{-1}$. The instability of most steep terrain is to be expected, as there is very little deep-rooted vegetation to prevent the unconsolidated material from sliding. In our field campaigns we observed that soil covered slopes, especially in the Niyaqu basin, feature Kobresia pygmea pastures, which forms a grass mat with a thick root system of up to 30 cm.

This may provide some stability in the absence of larger vegetation, however, both climate change and overgrazing are degrading this grass mat (Miehe et al., 2008), which could lead to larger sliding velocities in the future.

When studying relatively fast land surface changes with InSAR, it is important to consider the maximum LOS displacement that can be calculated reliably between two SAR acquisitions. Among

other factors this is dependent on the wavelength of the satellite (5.6 cm for Sentinel-1) and the temporal baseline of the interferogram. Measurements of displacement exceeding a quarter of the wavelength between two acquisitions are unreliable (Crosetto et al., 2016) and are likely to lead to an underestimation of the displacement signal and low coherence values. This is the case for some of our fast moving landforms during the summers of 2016, when the temporal baseline is up to 96

645  days for the Qugaqie basin in descending orbit. It is therefore likely that we underestimate velocities during that time period. Most interferograms feature much shorter baselines of 12 to 36 days and are therefore not affected by this issue. It is unlikely that this is the cause of the linear pattern of the fastest landforms, as the temporal baselines of the summer of 2018 are short and also do not show a clear acceleration of the velocity (Fig. 6B). Our data of Niyaqu basin is less affected by this

underestimation of surface velocity, as the maximum temporal baselines is 60 days and therefore only areas with a LOS velocity greater than 8.5 cm yr$^{-1}$ are affected. High velocity reduces coherence values in the center of the fastest landforms and leads to decorrelation in some cases.

We do not observe a clear seasonally accelerated pattern for most of the fastest moving landforms like rock glaciers. These landforms are creeping at comparatively linear rates, without distinct

differences between summer and winter (Fig. 6B), often with multiannual downslope velocities >50 mm yr$^{-1}$. We were able to identify 19 of these landforms in the Niyaqu basin and 33 in the Qugaqie basin by forming clusters of data points with a linear velocity pattern (less than 50% acceleration of the velocity in summer) and slope velocities >50 mm yr$^{-1}$. Our spatial data coverage of steep slopes is better in the Qugaqie basin compared to the Niyaqu basin. It is therefore unlikely that this 19 to 33

comparison is an accurate reflection of the difference in frequency of these landforms between both study areas. It is likely that some of these clusters have been misidentified as linearly moving, while actually featuring both the seasonal heave-subsidence cycle prevalent in the valleys and the seasonal

sliding pattern of the slopes. In some cases those two cycles may cancel each other out to such a degree that the resulting velocity appears linear. This can be observed at the interface between slopes and the valley (Fig. 7A/B).

We determined from optical satellite data, DEM analysis and field observations, that 36 % of these linearly creeping clusters are associated with rock glaciers or protalus ramparts, where motion is generally driven by massive ice within the landforms (Whalley and Azizi, 2003). Other studies observe strong seasonal variations in the velocities of rock glaciers (e.g. Kääb and Vollmer, 2000). Rock glacier kinematics are highly dependent on the climatic setting, ice content, ground lithology and slope (Haeberli et al., 2006), making comparison between rock glaciers of different regions difficult. Rock glaciers studied in north-western Bhutan show velocities of up to 300 mm yr$^{-1}$ and in rare cases up to 700 mm yr$^{-1}$ (Dini et al., 2019). However, neither study could analyze the seasonal displacement patterns of rock glaciers due to large temporal baselines of their interferograms. Strozzi et al. (2020) observe rock glacier velocities of approximately 1.5 to 2 m yr$^{-1}$ in the Argentinian Andes, 2 to 4 m yr$^{-1}$ in Western Greenland and 1 to 2 m yr$^{-1}$ in the Swiss Alps. The former two show a velocity increase of 30 to 50 % and the latter around 100 % between winter and late summer.

Not all fast and linearly moving areas are associated with landforms containing massive ice. Rock slope instabilities such as rock slides are common on the debris covered slopes and while most of them follow a seasonally accelerated displacement pattern, around 24 % of fast and linearly moving areas are likely associated with rock slope instabilities. We can therefore not be certain if fast linear motion is indeed an indicator of displacement driven by massive ice. Their relatively low dependency on seasonality indicates, however, that their displacement is mainly gravity-driven as opposed to slopes with strong seasonal variations, where the displacement is driven by both gravity and freeze-thaw related processes.

## 6.4 Seasonal displacements in steep terrain

Most slopes moving at least 10 mm yr$^{-1}$ experience a clear seasonal displacement signal, with velocities increasing considerably towards the end of the summer monsoon period (Fig. 6C). Monsoon season is associated with both the highest temperatures and approximately 80 % of the annual precipitation over a period of 4 months from June to September (NAMORS, 2018). For the Qugaqie basin it is also the only time when the average daily air temperature exceeds 0°C (Zhang et al., 2013). The clear connection between accelerated surface displacement and the increased air temperature in summer makes freeze-thaw related processes like solifluction a likely driver of displacements on soil covered slopes. Solifluction describes a process where seasonal freezing and thawing of the ground induces downslope displacement of up to 1 m yr$^{-1}$ (Matsuoka, 2001). Affected

slopes in the Qugaqie basin display downslope velocities of mostly 50 to 150 mm yr$^{-1}$ and up to 400 mm yr$^{-1}$ in some cases towards the end of the summer season.

## 7 Conclusion

Our InSAR time series analysis of Sentinel-1 data show clearly both multiannual and seasonal surface displacement patterns in the Nam Co area. Most flat areas are relatively stable on a multiannual scale but show a strong seasonal pattern induced by freezing of the active layer in late autumn and winter and its subsequent thawing in spring and summer. This induces a vertical oscillation with an amplitude of 5 to 10 mm in most regions with areas near water bodies showing a more pronounced

pattern with an amplitude of up to 24 mm. Most steep terrain in both study areas is unstable, due to the unconsolidated material and the lack of deep-rooted vegetation. They move downslope with velocities of 8 to 17 mm yr$^{-1}$. Most steep terrain also shows a seasonal displacement pattern driven by freeze-thaw processes, such as solifluction, on soil covered slopes and associated with rock slope instabilities, such as rockslides, on debris covered slopes. Downslope velocities on these slopes

accelerate from around 20 mm yr$^{-1}$ in winter to 50 to 150 mm yr$^{-1}$ in late summer for mean velocities of 30 to 70 mm yr$^{-1}$. The fastest landforms can reach mean velocities of 100 to 180 mm yr$^{-1}$. These landforms do not follow the seasonally accelerated sliding pattern of most slopes but creep linearly with little difference between summer and winter velocity indicating that they are gravity-driven. While we have identified some of those landforms as rock glaciers and protalus ramparts, we cannot

be certain to which extent fast linear velocity is an indicator for motion driven by massive ice in this area.

## Data availability

https://doi.pangaea.de/10.1594/PANGAEA.907743


## Team List

M. Sc. Eike Reinosch, Institute of Geodesy and Photogrammetry, Technische Universität Braunschweig, 38106, Braunschweig, Germany, e.reinosch@tu-braunschweig.de, phone: +49 531 391 94587

M. Sc. Johannes Buckel, Institute of Geophysics and extraterrestrial Physics, Technische Universität Braunschweig, 38106, Braunschweig, Germany, j.buckel@tu-bs.de, phone: +49 531 391 8506

Dr. Jie Dong, School of Remote Sensing and Information Engineering, Wuhan University, 430079, Wuhan, China, dongjie@whu.edu.cn, phone: +86 15072317889

Prof. Markus Gerke, Institute of Geodesy and Photogrammetry, Technische Universität
Braunschweig, 38106, Braunschweig, Germany, m.gerke@tu-bs.de, phone: +49 531 391 94570

Dr. Jussi Baade, Department of Geography, Friedrich-Schiller- Universität Jena, 07743, Jena, Germany, jussi.baade@uni-jena.de, phone: +49 3641 9-48803

Dr. Björn Riedel, Institute of Geodesy and Photogrammetry, Technische Universität Braunschweig, 38106, Braunschweig, Germany, b.riedel@tu-bs.de, phone: +49 531 391 94593


## Author Contribution

The majority of the scientific writing and the figures were produced by Eike Reinosch. He also performed most of the literature research and data processing. Johannes Buckel performed literature research of the study areas, especially about their geomorphology, and wrote parts of the
respective section. Furthermore he proof read the entire document regarding geomorphological and geological data and established connections between the results of the satellite analysis and relevant geomorphological landforms and processes in the field. Dr. Björn Riedel provided guidance regarding InSAR processing and proof reading of the manuscript, with a focus on the technical aspects of InSAR time series analysis. He also secured the funding for this research as part of the TransTiP project.
Prof. Markus Gerke proof read the manuscript, provided guidance about the relevant research questions, research direction and the manuscript structure and aided in establishing connections to other remote sensing instituting to discuss the content of this research with fellow researchers. Dr. Jussi Baade secured funding for the project, evaluated potential study areas and provided us with additional data through additional proposals to the DLR. Dr. Jie Dong performed a review of the
methods used with a focus on the seasonal displacement signal present in our data and the potential causes thereof.

## Competing Interests

The authors declare that they have no conflict of interests.


## Acknowledgements

We would like to thank the DLR for providing us with the high-resolution TanDEM-X DEM for our data processing (Proposal ID DEM_HYDR1727) and the ESA and Copernicus for making Sentinel-1 and Sentinel-2 data freely available to the public. We are especially grateful to the 'Deutsche
Forschungsgemeinschaft' (DFG) for funding our work as part of the Sino-German project 'Geo-ecosystems in transition on the Tibetan Plateau' (TransTiP; GRK 2309/1). We thank our Chinese colleagues, who have worked tirelessly to support us before and during our field work.

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
