# Peer review of "InSAR time series analysis of seasonal surface displacement dynamics on the Tibetan Plateau"

_The Cryosphere, 2019_

## Short Comment (SC1) · 15 Dec 2019

By Simon Daout and Benedetta Dini

We would like to make some comments on this publication, since we have been cited a few times. We believe that some interpretations have been discredited by the authors by providing incorrect statements. Moreover, we feel that some citation are made inappropriately (and incorrectly), denoting a superficial reading of exsisting literature.

l. 425. Other studies (Li et al., 2015; Daout et al., 2017) have stated that there is often a significant lag time between the day of maximum air temperature and the day

of maximum subsidence.

The day of maximum subsidence cannot be associated with the day of maximum temperature as it is perfectly known with in-situ ground monitoring and permafrost models that the active layer temperature does not follow a diffusive model but is mainly controlled by the Stefan equations (Riseborough, 1996). In other words, the subsidence has been shown to continue, at lower rates, well after the day of maximum temperature, until the temperature falls below zero. In situ-measurements (eg. Gruber et al., 2019 and many others) image this seasonal pattern, which can differ slightly from the Stefan model prediction depending on the moisture content, the snow coverage, the vegetation cover... In addition, the thawing of the ice-rich layers, together with the thaw settlement, can be delayed by few months from the freezing onset. For instance, Liu et al. (2017) document changes in active-layer thickness for the Tien Shan and show with their detailed time/depth graphs that complete active-layer refreezing at depth commonly takes place around the end of the year. Therefore, the lag time between the day of maximum air temperature and the day of maximum subsidence is not a statement from Daout et al., 2017 but a fact.

l. 433. We used this lag time to determine the active layer thickness (ALT) by assuming the heat transfer to be one-dimensional and the soil to be homogeneous.

The freezing onset is at first order controlled by the time at which surface ground temperatures drop below zero. Amplitude and timing of deformation are then controlled by the water/ice availability and the amount of excess ice in the ground (e.g Daout et al., 2017, Dini et al., 2019). It is, therefore, wrong to draw direct links between the observed deformation and the active layer thickness because the active layer does not follow a purely diffusive model and its behaviour in response to freeze-thaw is associated with the ability of the soil to retain water (grain size, mineralogy..) and the soil thickness.

l. 494. We observe a significantly shorter lag time, with most areas in this basin reaching their maximum subsidence ahead of the maximum air temperature by a few

days to weeks.

The absence of lag between the day of maximum air temperature and the day of maximum subsidence is most likely linked to a misinterpretation of uncorrected tropospheric delays which is instead attributed by the authors to freeze/thaw related processes. This is also supported by the clear correlation at high-frequency (i.e. well localised patterns following topography) and large scale between the seasonal amplitude and the topography (e.g Fig. 5). As Dini et al., 2019 (Remote Sensing of Environment) show, the attempt to remove atmospheric effects with the use of filters on the time series does not completely remove the layered atmosphere effects. For this reason, Dini et al. show that unless the interferograms are corrected before the time series generation, it is important to apply further corrections, such as those that use atmospheric models and/or empirical corrections generated by looking at the signal-topography correlation. In the aforementioned work, there are plenty of examples taken from a large scale study that indicate the important effects of such corrections and that show the atmospheric-dominated seasonal cycles before applying such corrections. The authors do say that they perform a linear spatial trend correction to mitigate for that, however it is not very clear what this involves and the homogenous timings of maximum subsidence look suspicious for non-atmospheric processes.

l. 486. We could not identify any significant difference in the freeze-thaw cycle between areas where permafrost is likely to be present and areas where the ground is only seasonally frozen. We therefore disagree with similar studies (Daout et al, 2017; Li et al., 2015) that associated this process with permafrost.

Frost heave/thaw settlement is primarily caused by the formation/thawing of excess ice (these depending on water content and porosity of the soil), especially through ice lenses formation (segregation ice) in frost-susceptible materials (silt, fine sand, loess) with high water content. Permafrost acts as an impermeable layer that retains the soil-moisture and isolates the active-layer from the deeper ground temperature gradient. Freeze/thaw cycles are therefore mainly detectable in permafrost regions, where the

soil contains enough ice/water content to produce thaw settlements higher than 0.5-1cm. In addition, it is evident that the point of change from subsidence to heave around October/November shown in Daout et al., 2017 relates to delayed thawing at depth (see comment 1), followed by heave as a consequence of the freezing and increasingly cold temperatures penetrating at depth until complete refreezing causes a period of winter inactivity. Also, large-scale models (e.g Qin et al., 2017, Gruber et al., 2012) have described the north-western part of the Tibetan plateau, studied in Daout et al., 2017, as a cold and continuous permafrost region with mean annual ground temperature below -5°C. Daout et al., 2017 can only describe permafrost related process and it is, therefore, unreasonable to think that the observed thaw settlement effects could be associated to freeze/thaw processes in non-permafrost areas.

l. 549. However, Dini et al. (2019) did not project their data along the steepest slope, which explains the lower values, and neither study analyzed the seasonal displacement patterns of rock glaciers in their study area.

The article that the authors incorrectly cite (rock glaciers velocities in Bhutan were analysed in Dini et al. 2019 published in RSE, not Dini et al., 2019 published in Engineering Geology) does indeed talk about rock glaciers velocities as they are projected on the steepest slope gradient. The method of assuming that for slope processes (i.e. landslides and rock glaciers) the velocity can be approximated to the steepest gradient is, in fact, quite well established. The authors present in this article a method to calculate a coefficient (correctly citing Notti et al., 2012) which was in fact generated in full by Notti et al. (2012). This is what is also applied in Dini et al. (2019, RSE). Citing from Dini et al., 2019: "If the displacement vector is assumed to be oriented downslope along the maximum gradient, which is a generally acceptable first assumption for gravitational slope movements, then it is possible to estimate the percentage of displacement detectable in the LOS (Notti et al., 2012) and thus to estimate a downslope velocity closer to the true velocity." In addition to this, Dini et al. (2019) looked for decorrelation over rock glaciers in their SBAS results. As the velocities were projected on the maximum

gradient and clear decorrelation corresponding to a rock glacier throughout the area of study was not found, it seems fair to state that the relatively slow movements observed over rock glaciers are real (at least over the observation period) and not an effect of misinterpretation of the InSAR results. In addition, the reason why Dini et al. 2019 have not analysed the potential of seasonal accelerations and deceleration of rock glacier movements is due to the temporal sampling of ENVISAT and ALOS, which is on average of 90 days, and therefore completely unsuitable to look at seasonal velocity variations.

Bibliography

- Daout, S., Doin, M.-P., Peltzer, G., Socquet, A. & Lasserre, C. Large scale InSAR monitoring of permafrost freeze-thaw cycles on the Tibetan Plateau. Geophys. Res. Lett. (2017).

- Dini et al., Classification of slope processes based on multitemporal DInSAR analyses in the Himalaya of NW Bhutan. Remote Sensing of the Environment, 233. (2019) Gruber, S. Derivation and analysis of a high-resolution estimate of global permafrost zonation. The Cryosphere, 6, 221–233 (2012)

- Gruber, S. Ground subsidence and heave over permafrost: hourly time series reveal inter- annual, seasonal and shorter-term movement caused by freezing, thawing and water move- ment. The cryosphere (2019).

- Liu, G. et al. Permafrost warming in the context of step-wise climate change in the tien shan mountains, china. Permafrost and Periglacial Processes 28, 130–139 (2017).

- Qin, Y. et al. Numerical modeling of the active layer thickness and permafrost thermal state across qinghai-Tibetan plateau. Journal of Geophysical Research: Atmospheres 122, 11–604 (2017).

- Riseborough, D., Shiklomanov, N., Etzelmuller, B., Gruber, S. & Marchenko, S. Recent advances in permafrost modelling. Permafrost and Periglacial Processes 19, 137–156
(2008).

---

## Author Comment (AC1) · 10 Jan 2020

Thank you very much for your insightful and helpful comments. We will respond to each comment in the same order as Daout and Dini.

I. We agree with your statement, that the lag time between maximum air temperature and maximum subsidence is a widely accepted fact, rather than a new statement made by Daout et al. (2017). We will change our manuscript accordingly, to make sure that this is clear to the reader.

II. The approach of Li et al., (2015) to determine the active layer thickness (ALT) from

the lag time between maximum air temperature and maximum subsidence is very sim-
plistic, especially as it does not consider variations in the ground moisture content. We
will therefore remove the section about the calculation of the ALT in the next draft of
our manuscript.

III. We do not agree with the statement that we misinterpreted the tropospheric delays
as freeze-thaw related processes. The reasoning of Daout and Dini to assume this
to be the case is (1) the seasonal patterns shown by us follow topographic structures
and (2) the correlation between the amplitude of the seasonal patterns and topogra-
phy. They also ask for clarification on our linear spatial trend correction, which we will
answer in (3).

(1) In our opinion Daout and Dini do not take into account, that we separated seasonal
freeze-thaw related processes into two different models: the freeze-thaw model in flat
areas / valleys and the seasonal sliding model on slopes (described in sections 4.4
and 4.5 respectively). It may therefore seem like our seasonal freeze-thaw related
processes follow topographic structures when looking at only one of these models.

(2) If the seasonal pattern we observe in our data was caused by tropospheric delay
and not ground deformation, then we would expect to see a correlation between the
strength (i.e. the amplitude) of this seasonal signal and the elevation. This has been
shown for example by Dong et al. (2019) or Dini et al. (2019). This is not the case in
our data (Fig. 1 below). We selected not only one reference point but instead 50 – 90
(Section 4.2), which should also help to reduce the effect of tropospheric delay on our
results.

(3) For linear spatial trend correction of the Qugaqie basin we used only regions we
expect to be relatively stable on a multiannual scale (i.e. flat and not in immediate
contact with water bodies or glaciers). We then determined the linear correlation of
their multiannual surface velocity and their elevation. The resulting linear trend ($R^2$ =
0.12 for ascending and $R^2$ = 0.38 for descending) was then removed from all ascending

and descending data points.

IV. We will change the section in question, as Daout and Dini point out correctly, that the comparison drawn in our manuscript is not appropriate here.

V. We agree that the publication Dini et al. (2019) in "Remote Sensing Environment" is a more suitable citation and we will change the sections in question accordingly.

References:

Daout, S., Doin, M. P., Peltzer, G., Socquet, A. and Lasserre, C.: Large‐scale InSAR monitoring of permafrost freeze‐thaw cycles on the Tibetan Plateau. Geophysical Research Letters, 44(2), 901-909, https://doi.org/10.1002/2016GL070781, 2017.

Dini, B., Daout, S., Manconi, A. and Loew, S.: Classification of slope processes based on multitemporal DInSAR analyses in the Himalaya of NW Bhutan. Remote Sensing of Environment, 233, 111408, https://doi.org/10.1016/j.rse.2019.111408, 2019.

Dong, J., Zhang, L., Liao, M. and Gong, J.: Improved correction of seasonal tropospheric delay in InSAR observations for landslide deformation monitoring. Remote Sensing of Environment, 233, 111370, https://doi.org/10.1016/j.rse.2019.111370, 2019.

Li, Z., Zhao, R., Hu, J., Wen, L., Feng, G., Zhang, Z. and Wang, Q.: InSAR analysis of surface deformation over permafrost to estimate active layer thickness based on one-dimensional heat transfer model of soils. Scientific Reports, 5, 15542, https://doi.org/10.1038/srep15542, 2015.

Fig. 1: Diagram of 1000 randomly selected data points (normalized for the lower frequency at higher elevations) showing the relationship between the amplitude of the seasonal signal and elevation.

[Figure]

R² = 0.0135

Fig. 1.

---

## Referee Comment (RC1) · Anonymous Referee #1 · 30 Jan 2020

An InSAR time series analysis considering a temporal interpolation of the displacement signals following various velocity models for areas where coherence is not maintained over a given threshold for all interferograms was applied to Sentinel-1 images over two locations around Nam Co Lake on the Tibetan Plateau. Results are used to study freeze-thaw processes, seasonal sliding and linear creep and are discussed with re-gard to the local geological knowledge and put in the context of the few other studies performed over the Tibetan Plateau.

The manuscript is well written and concise. The style chosen by the authors includes many statements with condensed information and very few general background infor-

mation. Much is written with the necessary approximation, but the statements are essentially correct in their formulation and content. I appreciate very much this style of presenting the work, even if one has to take into consideration the fact that a reader without a deep knowledge on many different aspects (InSAR, periglacial phenomena, etc.) might find it difficult at some points to follow the discussion.

The InSAR analysis includes many assumptions (e.g. regarding the interpolation when coherence is lost or the interpretation of the observations in the satellite line of sight direction), but this is well communicated. The images are prepared with great care and include a lot information, that the authors decided not to comment in every detail but rather to summarize for what they considered the most relevant aspects (which again I very much appreciate).

The paper is pertinent to The Cryosphere and I recommend minor revision with consideration of the following points.

l. 153. Remove exclusively.

l. 164. Please be more precise regarding the selection of the temporal and geometrical baselines. Which are minimum and maximum time intervals included in the analysis? As far as I know the Sentinel-1 baseline tube is consistently kept very small so that spatial decorrelation should not be an issue. Did you really exclude interferograms based on the spatial baseline?

l. 205. What exactly do you mean by "the orbital phase was corrected via a polynomial function"? Which function did you used? How did you determined the coefficients?

l. 214. The paper by Dong et al. (RSE, 2018, https://www.sciencedirect.com/science/article/abs/pii/S003442571930389X) might be of interest in this case and should be possibly included in the reference list.

l. 199. A coherence value of 0.1 is very low, really close to the pure noise level. If most of the interferegrams have in any case a much larger coherence value and the

0.1 threshold was considered to be able to have a spatially consistent solution, then I can understand this choice. But if most of the interferograms have such a low level of coherence, then the results would not be reliable. Please comment.

l. 310. Change the order of the columns of Table 2 to reflect the order of the discussion of Sections 4.3 to 4.5.

ll. 382-384. This is in my opinion too speculative and should be removed. As stated at lines 362 and 396 to 398, vegetation, streams and other water bodies are common phenomena which reduced the coherence without being to be considered "unstable".

Figure 3. Harmonize the order of the legends with panels A and B. Acknowledgements to Copernicus and DLR are not required here.

l. 259, ll. 531-532 and l. 571-572. A drawback of all InSAR time series techniques is the maximum detectable rate of motion, which is related to the possibility to correctly unwrap the phase. A phase cycle at C-band corresponds to 2.8 cm and aliasing are well possible already for half of that value. As mentioned before, the maximum time interval considered in your analyses is missing, but if interferograms spanning several months are considered, than I would expect problems in correctly computing the rate of motion already for few tens of cm/yr (e.g. for three months $2.8 Ãů90×365 = 11.35$ cm/yr). If small coherence values are retained, than the most obvious consequence of such an analysis is an underestimation of the rate of motion, in particular for small objects. Please consider if underestimation of the rate of motion for the most rapidly moving detected landforms cannot be the reason of not seeing a change of motion during the year. In addition, include a statement about what you estimate to be the maximum detectable rate of motion of your analyses.

---

## Author Comment (AC2) · 3 Feb 2020

Thank you very much for your helpful comments, we will reply to them in the same order.

I. 153: done

I. 164: The temporal baselines of our interferograms are 12 to 60 days for the Niyaqu basin and 12 to 72 and 12 to 96 days for the Qugaqie basin ascending and descending orbits respectively. Spatial baselines of Sentinel-1 are indeed small compared to other satellite systems and in most cases there are no problems. We discarded a small

number of interferograms with relatively long spatial baselines ( $\sim$ 200 m) due to poor coherence. We will remove the mention of spatial baselines from this section, as the low coherence rather than the spatial baselines of those interferograms were the deciding criteria. We attached the connections graphs of both study areas and both orbits to this comment (Fig. 1). We will adapt the relevant paragraph in our manuscript to describe the temporal baselines of our data sets and clarify our choice regarding the spatial baselines.

I. 205: Thank you for pointing this out, this paragraph is not correct. It should read: "the orbital phase was removed by subtracting a constant simulated phase from our interferograms. We then estimate a 3rd order polynomial function over flat stable areas and subtract this phase to remove any remaining large scale phase ramps."

I. 214: The publication linked is already part of our reference list. We cite it in I. 479.

I. 199: We attached a Figure (Fig. 2) to this comment showing the spatial coherence of our ascending data set over the Qugaqie basin (which has the highest percentage of low coherence data points). The low coherence brackets (0.1 to 0.2 and 0.2 to 0.3) are almost exclusively found near rivers and lakes and in this data set they represent 1.3 % and 3.0 % respectively. In our other data sets the data points with coherence values

both the fastest moving landform at up to 8 cm/yr in LOS and then longest temporal baselines (96 days in summer 2016 and 72 days in summer 2017). The coherence of this landform is 0.35 to 0.5. We attached a time series diagram of the cumulative surface displacement at this location (Fig. 3 top). Interferograms of 2018 feature temporal baselines of 12 to 36 days but also do not show seasonal variations in velocity. The longest temporal baseline of 96 days corresponds to a maximum surface velocity of 10.6 cm/yr. It is therefore likely that the velocities of the fastest landforms are being underestimated but not to a huge degree, as the temporal baselines are generally between 12 to 36 days. And only a small number of interferograms in the summer of 2016 and 2017 have long baselines when an underestimation of the displacement is likely. We do not believe that this is the cause of the lack of a seasonal signal in these landforms, as the time series of those landforms do not show an acceleration of their velocity in summer 2018 either. The temporal baselines of summer 2018 are 12-48 days. This is also corroborated by slightly slower landforms ( $\sim$ 4 cm/yr LOS) with similar surface characteristics as the fastest landforms, which also display this linear motion pattern but are less likely to suffer from an underestimation of the displacement signal. Figure 3 (bottom) displays the cumulative surface displacement of two different landforms with velocities  $\sim 4$  cm/yr LOS. The landform corresponding to the green line has similar surface characteristics as the fastest landforms and moves at a constant velocity, while the landform corresponding to the red line shows a strong seasonal signal. We will update the relevant paragraphs to show that we considered the possibility of underestimating the displacement signals of the fastest landforms.

**TCD**
*Figure 1:* SBAS connection graphs of ascending and descending data sets for both study areas. Data acquisition shown in red were discarded due to poor coherence of interferograms.

Fig. 1.

---

## Referee Comment (RC2) · Anonymous Referee #2 · 18 Feb 2020

The article from Reinosch et al. presents a case study of InSAR-measured displacements patterns related to freeze/thaw processes in two basins of the Nam Co area (Tibetan Plateau). The authors aim to study the seasonal and interannual dynamics of the ground in a periglacial environment. The objectives of the paper are relevant in term of research in SAR remote sensing applied in geosciences, and for understanding the short and long-term changes of periglacial processes at landscape-scale. In this way, the topic appears to be suitable for The Cryosphere. However, to my opinion, four major issues have to be addressed by the authors before the paper could be accepted for publication (see thereafter). There are also several complementary elements that should be considered (listed at the end of the review).

Note that I decided not to read the others comments posted during the open review process in order to avoid being influenced. There are thus potentially repetitions with the reviewer 1 and other persons who previously commented on the article.

...

—Major comments—

...

—1. Confusion between environmental factors, processes and InSAR-measured effects—

The air temperature (environmental factor) transferred into the ground and varying under and over zero degree leads to phase change of water/ice (process), that leads to frost heave and thaw subsidence (effect). Of course, there is a link between these elements but it is misleading to present InSAR as a technique able to directly measure the freeze/thaw cycles and thermal properties of the ground. In addition, subsidence, even in periglacial environment, can be measured without being necessarily related to thaw. This confusion is present all along the manuscript. Here an non-exhaustive list of examples:

- l.17 and l.51: "surface displacement processes", "seasonal displacement processes": use "processes" or "surface displacements" separately. A surface displacement is the consequence of a process.

- l.31: "We observe a very clear seasonal freeze-thaw cycle": you do not observe the freeze-thaw cycle, you observe heave and subsidence.

- l.311: "freeze-thaw model"

- l.318: "freeze-thaw amplitude"

- Fig.5: "freeze-thaw amplitude" (or "freezing and thawing parameters" in the legend) is not the right terminology, you measure heave/subsidence amplitude. "Maximum

freeze-thaw subsidence" should also be replaced by "maximum thaw subsidence"

- l.406-407: "The day of maximum subsidence (DMS) is the day in summer during which the soil has thawed to its maximum extent". For the same thaw depth, subsidence can vary depending on the ice content.

On slopes, it is important to think the same way: various processes can lead to an apparently similar effect (downslope movement). See major comment 2.

...

—2. The three models, the downslope projection and the assumption of variable ice content to differentiate linear/seasonal patterns on slopes—

I would suggest to consider other names for the models and be more clear about their differences from the start (before 4.3). A summary comes at l.333-338, but it is a bit too late. It would be easier to follow if the overall idea is clearly explained just before l.257. FTM name could be changed to heave/subsidence for the reason explained in major comment 1. SSM name is not fully correct: the landforms may have seasonal acceleration/deceleration but do not fully stop creeping. MSM is in general not clear to me: in areas <10deg, what is the difference with FTM? Did you remove the seasonal trend to keep only the multi-annual trend? Is SSM also based on projected results (not clearly stated at l.322-328)? Overall the names are mixing displacement patterns and related processes: maybe easier to choose either process-based names: for ex "heave-subsidence model", "seasonal slope process model", "linear slope process model" or displacement-based names: "vertical cyclic model", "downslope cyclic model", "downslope linear model" (just as examples).

The assumption of projection along slopes, if mostly right for landslides and rock glaciers, can be problematic for processes including both downslope and heave/subsidence components (such as solifluction,with displacement normal to slope in winter and vertically down in summer). In addition, it has been documented that
these processes can occur on slopes <10 deg (see Matsuoka, 2001). I understand the need to simply but this limitation should at least be acknowledged in the manuscript. Gravity-driven downslope pattern does not necessarily mean permafrost creep, even in periglacial environment. Have you considered the potential presence of rock slope instabilities in these areas? If it sounds possible, you could be a bit more modest in the assumption relating linearity with high ice content (as you state at l.457-462; l.540-543). If not likely in these areas, it has to be stated.

Overall, maybe consider to use "downslope-dominated" (or gravity-driven) vs "vertically-dominated" (or freeze/thaw-driven)Âż instead of speaking about linearity/seasonality. As you write at l.536-538 (too late and too little explained to my opinion), it can "have been misidentified as linearly moving, while actually featuring both the seasonal freeze-thaw cycle prevalent in the valleys and the seasonal sliding pattern of the slopes". A way to check this it to plot the velocity in addition to (or instead of) the cumulated displacement on Fig.6. Fig.6B may look linear but looking at the velocity, I think you may see variations as well.

...

—3. SAR data, ISBAS processing, interpretation of uplift areas and DMS—

Information about several basic data properties and methodological information (important for interpreting the results) is missing: multilooking factor, final spatial resolution, LOS angles, spatial/temporal baseline thresholds, temporal distribution of the initial SAR scenes (baseline plot), map with coherence, map with location of reference areas (in supplementary material for ex).

About ISBAS processing: l.189-190: "... where the coherence is intermittently below the chosen threshold..." and l.198 "... near water bodies, where coherence is very low. We therefore decided to use a very low coherence threshold of 0.1 to increase...": I am not especially known with ISBAS approach, but this sounds quite dangerous to me, especially if you used a threshold of 0.1 in some areas (l.198). Does it mean that

you have 25% of interferograms with <0.1 in these areas? Maximizing the coverage also to areas where the results cannot be reliable due e.g. to vegetation or moisture means that some of your interpretation can be based on wrong estimates. At least good to try to explain as much as possible the method, document the uncertainties and acknowledge the potential limitations (in methods and/or in discussion). A coherence map could also be a nice way to document the distribution of these Âńless reliableÂż areas.

Due to this lack of information, it is hard to fully understand the cause of the uplift detected in some flat valley bottoms (l.366 and Fig.3). Looking also at Fig.6A, if you subtract the last and the first acquisitions, you also get a positive trend. Is it really likely that all these locations are affected by sediment accumulation or can it be a bias? I wonder if this cannot be due to low reliability (low coherence) in these areas, especially during the "wet" periods when the ground is subsiding. Or a bias due to the temporal sampling of the initial interferograms? Or atmospheric effect (remaining stratified component)?

About DMS: at l.413-414, it is written that there is shift of 11-27 days between ascending/descending datasets. Why that? 11/27 days is quite a lot, considering that it should in theory document the same thing. Can it be due to a shift of velocity value (problem with the location of the reference points?) or the different LOS incidence angles (different sensitivity to the vertical)? Due to undocumented information about data properties, it is hard to understand the results and fully trust them.

...

—4. Poor discussion, bold/vague statements—

DMS 9 days prior to the temperature peak in one of the AOI is presented as "no lag" (l.422, l.427): this is in fact an inverse lag (or lag in the "wrong" direction), which has to be discussed. I would guess this may be due to the distance to the meteorological station: NAMORS station is maybe not representative of this AOI considering that

Qugapie has a significantly higher elevation? Did you try to apply an altitude correction? Figure 3: The bottom of the graph E/F is too little explained/exploited in the manuscript. To my opinion, this is maybe the most interesting finding of the study. There is a lack of structure in the Discussion. Consider dividing the Section in three parts, for ex: "Uncertainties/Error source"; "Thaw subsidence / Frost heave cycles"; "Downslope processes".

The Discussion is also quite vague. Some examples thereafter:

- l.480-481: one major error source in periglacial environment (during summer) is the impact of ground moisture on the phase (moisture can lead to a biased detection of distance change, up to 10-20% of the wavelength). Good to discuss this. See e.g. references: De Zan et al., 2014; Zwieback et al., 2017.

- l.485-486: "it does not follow the trend of a sine curve perfectly. Nonetheless we consider it a valid if imperfect approach". Really vague.

- l.486-488: "not identify any significant difference in the freeze-thaw cycle between areas where permafrost is likely to be present and areas where the ground is only seasonal frozen": this has not been presented in the Results.

- l.488-489: "we therefore disagree with similar studies..." without explaining more about the results and differences with the other studies, it is a bit too bold to say this...

- l.489-491: "the amplitude, the day of maximum thaw subsidence and the active layer thickness... agree well": summarize the values and explain more.

- l.498: "the point density of their data within Qugaqie basin is too low to draw reliable conclusion": I would also say that the meteorological data you have available may also be too weak to draw reliable conclusions.

- l.499-500: "explained by the small size of the basin": why the size would have something to do with the lag time? Have you considered other explanations: topography/location/altitude/permafrost extent? Looking at Fig.1, it appears that half of

Qugaqie basin is in permafrost zone, while Niyaqu basin is mostly in seasonally-frozen ground. These differences (and the consequences on your results) could be discussed.

- l.524-525: I believe you write here a start of answer to my previous question.

- l.511-512: How would be carefully speaking about permafrost degradation with only 3 seasons.

- l.543: "This disagrees...". It is not because you did not find strong seasonal variations, that your study disagrees with the study you refer to. Not the same context, focus, method, etc. And written at l.534-538: it can have both a downslope and a seasonal pattern. As it is not based on 3D measurements and projection has been performed, we cannot know.

...

—Complementary comments—

—Abstract—

It could follow a better structure (context, processes you aim to study, presentation of the data and methods, and main results/conclusions).

- l.19: "these processes": not clear which processes . Last lines of the abstract explaining the active layer freeze/thaw and related displacements could come earlier.

- l.21: "Sentinel-1 constellation" or "Sentinel-1 satellites"

- l.25: "monsoonal climate accelerates those movements". Confusing as you say just after that not all are affected by seasonal variations.

- l.30-31: "permafrost degradation": I would be careful drawing conclusions about permafrost degradation based on only 3 seasons.

—1. Introduction—

The introduction is overall a bit poor. It currently focuses a lot on the Tibetan Plateau,

it could benefit for other references to similar kind of studies in others regions of the world. Here an non-exhausive list of ref. that could also contribute to go further with the discussion of your findings: in Alaska: Liu et al., 2010, 2012; Schaefer et al., 2015; in Canada: Short et al., 2014; Rudy et al., 2018; in Greenland: Strozzi et al., 2018; in Svalbard: Rouyet et al., 2019; in Siberia: Antanova et al., 2018.

- l.45: "...harder to quantify using optical remote sensing due to their debris cover". I do not see what is the problem of the debris cover. I would say that the major reasons that there are difficulties and lack of inventories are: 1) small features, difficult to document if the spatial resolution is poor, and potentially slower than glaciers (problematic for optical remote sensing techniques); 2) more research focus on glaciers due to size/impact. Mountain permafrost is a relatively new field. But some studies using remote sensing are available, see e.g.: Kellerer-Pirklbauer et al., 2012; Millar & Westfall, 2008, Rangecroft et al., 2014; Lilleøren & Etzelmüller, 2011; Barboux et al., 2015.

- l.46: "...despite their importance as water storages". I would add: the evidence of a link between rock glacier kinematics and climate change. See e.g.: Delaloye et al., 2010; Kääb et al., 2007; Kellerer-Pirklbauer & Kaufmann, 2012; Roer et al., 2005; Ikeda et al., 2008; Kenner & Magnusson, 2017.

- l.52: inverse the order -> there is no thaw subsidence before a previous frost heave.

- l.54: "InSAR results" instead of "models"

- l.57-58: Rephrase. Maybe: "SAR satellites are side-looking and observe the Earth obliquely."

- l.65-67: this is more for "Methods" than "Introduction". I would simplify here (for ex just say "lead to poor phase stability, so-called coherence"). And go further with it in Section 4.

- l.70: "This is not a problem in our study sites.... In fact, we found..." It does not really fit in "Introduction". Corresponds more to "Results". And without further explanation

(dry-little precipitation -> little snow cover), it is a bit useless.

- l.76-77: "rising lake level": this has not much to do with your study. Could come as a general info about the context (first paragraph) but does not really fit here.

- l.82: As explained in major comment 2, the linearity is not in itself a proof for ice content. But it shows that the landform is gravity-driven (dominated by a downslope component). The assumption should at least be further discussed (in Section 6 for ex): What does "high ice content" mean? What about rockslides? What about solifluction processes (with ice content and both seasonal variability and downslope component)?

- l.85: "landforms without ice": without massive ice? Without significant ice content. Seasonal pattern does not mean "no ice".

—2. Study Area—

- l.95: rephrase "endorheic catchment borders on the catchments..."

- l.96-98: rephrase. For ex: "They have elevations up to 7162 m a.s.l. The highest parts are glaciated, while the other areas are considered to be in the periglacial zone."

- l.112 and l.114: "... with high vegetation, such as..."; "... lack of high vegetation..."

- l.117: "... due to significant change of physical surface characteristics..."

- l.129: mixing spatial/temporal limitations. For ex replace by: "... making continuous temporal coverage of..."

- l.128-129: rephrase "... different levels of glacial impact and the predominant landscapes and their related surface motion processes at Nam Co."

- l.138: "Other landforms were accumulated through ... processes". Material (sediment) accumulates and processes shape landforms. Not fully correct to say that a landform accumulates through a process.

- l.141: "virtually free of vegetation". What does "virtually" mean?

- l.140-143: What about the bedrock? Missing information about geological context.

—3. Data—

- l.156: what do you mean by unreliable? Or "stable" (l.158)?

- l.162-164: the part about the interferogramanalysis, baseline and coherence thresholds is already about the "Methods" (how the dataset has been processed). Move to Section 4 + important to document the chosen thresholds (max temporal/baseline baselines). The max temporal baseline determines the max. velocity you can detect.

- Table 1: I am missing info about the line-of-sight (important for ex. to know what is this incidence angle and thus the sensitivity to vertical/horizontal component for both geometries) + We don't know the temporal distribution of the SAR scenes (fully continuous? Some gaps?). A baseline plot (potentially in supplementary material) would be valuable.

—4. Methods—

- l.183-184: "with a short temporal baseline": the threshold (max. temporal baseline) is not documented. Important to add this and explain why it has been chosen (in relation with the detection capability and the max. expected velocity)

- l.189-190: "... where the coherence is intermittently below the chosen threshold..." and l.198: "... near water bodies, where coherence is very low. We therefore decided to use a very low coherence threshold of 0.1 to increase..." = 0.75 intermittent value with a 0.1 threshold applied (25% of interferograms with <0.1)? Sounds really liberal to me, good to discuss it (see major comment 3).

- l.209: "strong shift": shift or spatial trend?

- l.215-216: "the error range of the slope projection..." the sentence is not understandable here (before having explained how the projection is performed)

- l.233: "...must not have any significant velocity by themselves". Rephrase, no clear.

Don't you just mean that the points have to be in areas supposed to be stable?

- l.251: "LOS shifts": what does it mean? "velocity shifts along the LOS" maybe?

- l.273: "lateral sliding" is a bit counter-intuitive terminology. Horizontal component would be easier to understand.

- l.279-280: "exception for areas with a large east-west velocity". Be clear about how you selected these areas (which velocity threshold? Manual selection based on prior knowledge?). How to be sure you didn't miss any area <10 with a significant horizontal component.

- l.290-291: why smoothing? Without knowing the resolution of the final InSAR results, hard to understand.

- Table 2: MSM displacement type: not only along slope if I understood correctly, also vertical on flat areas. Related geomorphological processes: maybe be a bit more specific – permafrost creep can also have seasonal variations. FTM slope: what does Âńmostly <10degÂż means?

- l.320: here you speak about the resolution, it comes too late (as I guess it applies also for the other models) and there is no info about the multi-looked resolution before interpolation. You could also add an explanation about why it is important to interpolate (sounds unnecessary to me).

—5. Results—

- l.374: again here, prefer "horizontally" instead of "laterally"

- l.359-360: here some causes of temporal decorrelation are presented, but one is not explained clearly: too fast movement. This is also why it is important to document somewhere the max. temporal baseline used for InSAR processing.

- l.380-382: "most of the low coherence areas would likely be considered as unstable": I don't think it is right to say this: as written at l.359, the gaps in coverage can also

be due to shadow/layover, and in addition, low coherence can also be due to ground moisture.

- Fig.3: Due to the chosen color scale, it is really hard to see the difference between areas with vertical assumption or those with downslope projection. Also hard to spot the areas affected by subsidence. + Maps C and D are only for Qugaqie basin. Why? + I may have missed sth, but I think there is no mention of the "seasonal sliding coefficient" threshold used to differentiate "linear velocity" and "faster in summer" in map C.

- l.406-407: see major comment 1: Thaw subsidence is related to temperature, but is not a thermal property.

- l.413: "DMS of the descending dataset occurs earlier": why? (see major comment 3).

- l.449: "while the latter arrests most sliding processes". Âńslows downÂż would be more correct.

- Fig.6 "areas throughout Qugaqie basin": what about the other basin? As for Figure 3, not clear why you let one basin out of the analysis. + The locations of the selected points could be shown somewhere, on a map.

—6/7. Discussion/Conclusion—

(Mostly developed in major comment 4)

- l.518: missing a reference here.

- l.530-531: "by clustering data point with a strong linear pattern and high slope velocities": not clear what it means? Maybe add a map and some examples (InSAR vs orthophoto) in supplementary?

---

## Author Comment (AC3) · 20 Feb 2020

Thank you very much for your incredibly detailed and insightful comment to our manuscript. Due to the large amount of comments, we will not reply to each individual comment but instead will focus on the major comments and topics.

Major comments:

1. We agree that the terminology we use in the manuscript is at times confusing and does not differentiate well between displacements detected with InSAR and the underlying processes. We will adapt the terminology in our manuscript (based on your

suggestions) to make this clearer.

2. We will adapt the descriptions of our three models and their names to make it clearer where each model is applied, what their respective focus is, which processes are covered by each model and go into greater depth to assess their limitations in the discussion as you suggest.

a. Regarding your comment about "rock slope instabilities" and the value of submitting a supplement alongside our manuscript: We acknowledge that not discussing rock slope instabilities and solifluction in our manuscript was a grave oversight and we will adapt our manuscript to include them as potential causes of the observed displacement. The landforms with a high linear velocities are mainly rock glaciers, protalus ramparts and collapsing moraines. Rock slope instabilities are present throughout the study sites but in those areas we observe mainly seasonally accelerated sliding.

b. We agree that supplementary material would be very helpful to assess the quality of our data. Data we plan to include in the supplement are: - coherence and interferogram percentage maps of both study sites and both orbits (including the locations of the reference points) - baseline plots of our data sets, the maps of the seasonal sliding model of the Niyaqu basin - a map with the location of the time series shown in Figure 6 of our manuscript and their velocity time series (Fig. 1 of this reply) - maps showing the spatial distribution of the disparity between the "day of maximum subsidence" of ascending and descending data sets.

c. As you suggested we plotted the velocity time series of the locations shown in Figure 6 of our manuscript (Fig. 1 of this reply). You are correct, that the landforms shown in 6B of our manuscript (shown in black in Fig 1 of this reply) also show variations in their velocity. The seasonal variations of the velocity of the landforms shown in 6C of our manuscript (shown in grey in Fig. 1 of this reply) are however much larger. In our opinion plotting the cumulative displacements shows this seasonal behavior better than the velocity time series. We agree that it is important to acknowledge in our
manuscript, that the landforms we describe as moving linearly also show variations in their velocities, albeit smaller than other landforms. We will also add the velocity time series to our supplement.

3. The majority of this comment has been answered in the reply to the previous referee comment so we will summarize our reply. We will provide coherence maps and baseline plots in the supplement as you suggested and we added the additional information (multilook factors, incidence angle etc.) to our manuscript. We agree that the maximum detectable velocity is an important point do discuss and we added a paragraph to the discussion of our manuscript. We agree that increased soil moisture in summer could explain the supposed uplift in the valley of Qugaqie basin. We will explore that option in our discussion. We added a paragraph discussing the reliability of the day of maximum subsidence and address the disparity between ascending and descending results. We will also add a map of the spatial distribution of this disparity to the supplement.

4. Thank you very much for your assessment and very helpful comments of the discussion section of our manuscript. We adapted the discussion according to your suggestions.

Complementary comments:

1. Introduction: We agree that our manuscript will benefit from presenting studies of other regions besides the Tibetan Plateau in the introduction. We will update our manuscript accordingly.

2. Minor comments of our study areas, data and methods: We made the suggested changes.

3. Results: The reason why the slopes of Niyaqu basin are not covered in such great detail compared to Qugaqie basin (not included in Figures 3 and 6 of our manuscript) is that the spatial coverage of our InSAR data on the periglacial slopes of Niyaqu basin is much poorer. We added a paragraph to our manuscript to explain this. We will include

the map of the seasonal sliding model of Niyaqu basin in our supplement.

[Figure]

[Figure]

**Figure 1:** *Timeseries of LOS velocities of landforms shown in Fig. 6 of the manuscript. Negative velocity values indicate motion away from the satellite. Black time series refer to landforms with linear velocity (Fig. 6B in the manuscript) and grey time series refer to landforms with seasonally accelerated velocities (Fig. 6C in the manuscript). All timeseries represent moving averages of the 2 nearest values forwards and backwards in time.*

**Fig. 1.**

---

## Author Response (AR1)

**Reply to comments (comments in cursive with our reply below)**

**1. Replies to comments made by Daout and Dini**

20

a) The day of maximum subsidence cannot be associated with the day of maximum temperature as it 5 is perfectly known with in-situ ground monitoring and permafrost models that the active layer temperature does not follow a diffusive model but is mainly controlled by the Stefan equations (Riseborough, 1996). In other words, the subsidence has been shown to continue, at lower rates, well after the day of maximum temperature, until the temperature falls below zero. In situ-measurements (eg. Gruber et al., 2019 and many others) image this seasonal pattern, which can differ slightly from

10 the Stefan model prediction depending on the moisture content, the snow coverage, the vegetation cover... In addition, the thawing of the ice-rich layers, together with the thaw settlement, can be delayed by few months from the freezing onset. For instance, Liu et al. (2017) document changes in active-layer thickness for the Tien Shan and show with their detailed time/depth graphs that complete active-layer refreezing at depth commonly takes place around the end of the year.

15 Therefore, the lag time between the day of maximum air temperature and the day of maximum subsidence is not a statement from Daout et al., 2017 but a fact.

We agree with your statement, that the lag time between maximum air temperature and maximum subsidence is a widely accepted fact, rather than a new statement made by Daout et al. (2017). We changed our manuscript accordingly, to make sure that this is clear to the reader.

b) The freezing onset is at first order controlled by the time at which surface ground temperatures drop below zero. Amplitude and timing of deformation are then controlled by the water/ice availability and the amount of excess ice in the ground (e.g Daout et al., 2017, Dini et al., 2019). It is, therefore, wrong to draw direct links between the observed deformation and the active layer thickness because the active layer does not follow a purely diffusive model and its behaviour in

- 25 response to freeze-thaw is associated with the ability of the soil to retain water (grain size, mineralogy..) and the soil thickness.
- The approach of Li et al., (2015) to determine the active layer thickness (ALT) from the lag time 30 between maximum air temperature and maximum subsidence is very simplistic, especially as it does not consider variations in the ground moisture content. We therefore removed the section about the calculation of the ALT in this draft of our manuscript.

c) The absence of lag between the day of maximum air temperature and the day of maximum subsidence is most likely linked to a misinterpretation of uncorrected tropospheric delays which is 35 instead attributed by the authors to freeze/thaw related processes. This is also supported by the clear correlation at high-frequency (i.e. well localised patterns following topography) and large scale between the seasonal amplitude and the topography (e.g Fig. 5). As Dini et al., 2019 (Remote Sensing of Environment) show, the attempt to remove atmospheric effects with the use of filters on the time series does not completely remove the layered atmosphere effects. For this reason, Dini et al. show

40 that unless the interferograms are corrected before the time series generation, it is important to apply further corrections, such as those that use atmospheric models and/or empirical corrections generated by looking at the signal-topography correlation. In the aforementioned work, there are plenty of examples taken from a large scale study that indicate the important effects of such corrections and that show the atmospheric dominated seasonal cycles before applying such 45

that, however it is not very clear what this involves and the homogenous timings of maximum subsidence look suspicious for non-atmospheric processes.

We do not agree with the statement that we misinterpreted the tropospheric delays as freeze-thaw 50 related processes. The reasoning of Daout and Dini to assume this to be the case is (1) the seasonal patterns shown by us follow topographic structures and (2) the correlation between the amplitude of the seasonal patterns and topography. They also ask for clarification on our linear spatial trend correction, which we will answer in (3).

- (1) In our opinion Daout and Dini do not take into account, that we separated seasonal freeze-thaw related processes into two different models: the freeze-thaw model in flat areas / valleys and the seasonal sliding model on slopes (described in sections 4.4 and 4.5 respectively). It may therefore seem like our seasonal freeze-thaw related processes follow topographic structures when looking at only one of these models.
- (2) If the seasonal pattern we observe in our data was caused by tropospheric delay and not ground deformation, then we would expect to see a correlation between the strength (i.e. the amplitude) of this seasonal signal and the elevation. This has been shown for example by Dong et al. (2019) or Dini et al. (2019). This is not the case in our data (Fig. 1 below). We selected not only one reference point but instead 50 - 90(Section 4.2), which should also help to reduce the effect of tropospheric delay on our results.
  - (3) For linear spatial trend correction of the Qugaqie basin we used only regions we expect to be relatively stable on a multiannual scale (i.e. flat and not in immediate contact with water bodies or glaciers). We then determined the linear correlation of their multiannual surface velocity and their elevation. The resulting linear trend (R2 = 0.12 for ascending and  $R^2 = 0.38$  for descending) was then removed from all ascending and descending data points.

Fig. 1: Diagram of 1000 randomly selected data points (normalized for the lower frequency at higher elevations) showing the relationship between the amplitude of the seasonal signal and elevation.

d) Frost heave/thaw settlement is primarily caused by the formation/thawing of excess ice (these 75 depending on water content and porosity of the soil), especially through ice lenses formation

65

55

(segregation ice) in frost-susceptible materials (silt, fine sand, loess) with high water content. Permafrost acts as an impermeable layer that retains the soilmoisture and isolates the active-layer from the deeper ground temperature gradient. Freeze/thaw cycles are therefore mainly detectable in

- 80 permafrost regions, where the soil contains enough ice/water content to produce thaw settlements higher than 0.5- 1cm. In addition, it is evident that the point of change from subsidence to heave around October/November shown in Daout et al., 2017 relates to delayed thawing at depth (see comment 1), followed by heave as a consequence of the freezing and increasingly cold temperatures penetrating at depth until complete refreezing causes a period of winter inactivity. Also, large-scale
- 85 models (e.g Qin et al., 2017, Gruber et al., 2012) have described the north-western part of the Tibetan plateau, studied in Daout et al., 2017, as a cold and continuous permafrost region with mean annual ground temperature below -5\_C. Daout et al., 2017 can only describe permafrost related process and it is, therefore, unreasonable to think that the observed thaw settlement effects could be associated to freeze/thaw processes in non-permafrost areas.

90

We changed the section in question, as Daout and Dini point out correctly, that the comparison drawn in our manuscript is not appropriate here.

e) The article that the authors incorrectly cite (rock glaciers velocities in Bhutan were analysed in Dini et al. 2019 published in RSE, not Dini et al., 2019 published in Engineering Geology) does indeed talk
about rock glaciers velocities as they are projected on the steepest slope gradient. The method of assuming that for slope processes (i.e. landslides and rock glaciers) the velocity can be approximated to the steepest gradient is, in fact, quite well established. The authors present in this article a method to calculate a coefficient (correctly citing Notti et al., 2012) which was in fact generated in full by Notti et al. (2012). This is what is also applied in Dini et al. (2019, RSE). Citing from Dini et al., 2019:

- 100 "If the displacement vector is assumed to be oriented downslope along the maximum gradient, which is a generally acceptable first assumption for gravitational slope movements, then it is possible to estimate the percentage of displacement detectable in the LOS (Notti et al., 2012) and thus to estimate a downslope velocity closer to the true velocity." In addition to this, Dini et al. (2019) looked for decorrelation over rock glaciers in their SBAS results. As the velocities were projected on the
- 105 maximum gradient and clear decorrelation corresponding to a rock glacier throughout the area of study was not found, it seems fair to state that the relatively slow movements observed over rock glaciers are real (at least over the observation period) and not an effect of misinterpretation of the INSAR results. In addition, the reason why Dini et al. 2019 have not analysed the potential of seasonal accelerations and deceleration of rock glacier movements is due to the temporal sampling of ENVISAT
- 110 and ALOS, which is on average of 90 days, and therefore completely unsuitable to look at seasonal velocity variations.

We agree that the publication Dini et al. (2019) in "Remote Sensing Environment" is a more suitable citation and we changed the sections in question accordingly.

**115**

**2. Replies to referee comment 1**

Short comments referring to individual words or very minor changes are not listed here but we followed the suggestions of the referee and adopted all the suggested changes.

120 a) Please be more precise regarding the selection of the temporal and geometrical baselines. Which are minimum and maximum time intervals included in the analysis? As far as I know the Sentinel-1 baseline tube is consistently kept very small so that spatial decorrelation should not be an issue. Did you really exclude interferograms based on the spatial baseline?

- 125 The temporal baselines of our interferograms are 12 to 60 days for the Niyaqu basin and 12 to 72 and 12 to 96 days for the Qugaqie basin ascending and descending orbits respectively. Spatial baselines of Sentinel-1 are indeed small compared to other satellite systems and in most cases there are no problems. We discarded a small number of interferograms with relatively long spatial baselines (~200 m) due to poor coherence. We removed the mention of spatial baselines from this section, as the
- 130 low coherence rather than the spatial baselines of those interferograms were the deciding criteria. We adapted the relevant paragraph in our manuscript to describe the temporal baselines of our data sets and clarify our choice regarding the spatial baselines. We included the connections graphs of both study areas and both orbits in the supplement.

b) What exactly do you mean by "the orbital phase was corrected via a polynomial function"? Which function did you used? How did you determined the coefficients?

Thank you for pointing this out, this paragraph is not correct. It should read: "the orbital phase was removed by subtracting a constant simulated phase from our interferograms. We then estimate a 3rd order polynomial function over flat stable areas and subtract this phase to remove any remaining large scale phase ramps."

c) The paper by Dong et al. (RSE, 2018, https://www.sciencedirect.com/science/ article/abs/pii/S003442571930389X) might be of interest in this case and should be possibly included in the reference list.

145

135

140

The publication linked is already part of our reference list.

d) A coherence value of 0.1 is very low, really close to the pure noise level. If most of the interferegrams have in any case a much larger coherence value and the 0.1 threshold was considered
to be able to have a spatially consistent solution, then I can understand this choice. But if most of the interferograms have such a low level of coherence, then the results would not be reliable. Please comment.

Data points with coherence values

**Figure 2:** Cumulative LOS displacements of the fastest landform at ~8 cm/yr (top) and two different landforms with velocities of ~4 cm/yr (bottom) in Qugaqie basin in descending orbit.

**200 3. Replies to referee comment 2**

Short comments referring to individual words or very minor changes are not listed here but we followed the suggestions of the referee and adopted all the suggested changes.

a) The air temperature (environmental factor) transferred into the ground and varying under and over
 zero degree leads to phase change of water/ice (process), that leads to frost heave and thaw subsidence (effect). Of course, there is a link between these elements but it is misleading to present
 InSAR as a technique able to directly measure the freeze/thaw cycles and thermal properties of the ground. In addition, subsidence, even in periglacial environment, can be measured without being necessarily related to thaw. This confusion is present all along the manuscript.

210

230

We agree that the terminology we use in the manuscript is at times confusing and does not differentiate well between displacements detected with InSAR and the underlying processes. We adapted the terminology in our manuscript (based on your suggestions) to make this clearer.

b) I would suggest to consider other names for the models and be more clear about their differences
from the start (before 4.3). A summary comes at I.333-338, but it is a bit too late. It would be easier to follow if the overall idea is clearly explained just before I.257. FTM name could be changed to heave/subsidence for the reason explained in major comment 1. SSM name is not fully correct: the landforms may have seasonal acceleration/deceleration but do not fully stop creeping. MSM is in general not clear to me: in areas <10deg, what is the difference with FTM? Did you remove the</li>

220 seasonal trend to keep only the multi-annual trend? Is SSM also based on projected results (not clearly stated at I.322-328)? Overall the names are mixing displacement patterns and related processes: maybe easier to choose either process-based names: for ex "heave-subsidence model", "seasonal slope process model", "linear slope process

model" or displacement-based names: "vertical cyclic model", "downslope cyclic model", "downslope 225 linear model" (just as examples).

We adapted the descriptions of our three models and their names to make it clearer where each model is applied, what their respective focus is, which processes are covered by each model and go into greater depth to assess their limitations in the discussion as you suggest. We changed FTM (now called heave-subsidence model according to your suggestion) to only cover areas with a slope <10° to make it clearer where each model is applied.

c) The assumption of projection along slopes, if mostly right for landslides and rock glaciers, can be problematic for processes including both downslope and heave/subsidence components (such as solifluction, with displacement normal to slope in winter and vertically down in summer). In addition,

- it has been documented that these processes can occur on slopes <10 deg (see Matsuoka, 2001). I understand the need to simply but this limitation should at least be acknowledged in the manuscript. Gravity-driven downslope pattern does not necessarily mean permafrost creep, even in periglacial environment. Have you considered the potential presence of rock slope instabilities in these areas? If it sounds possible, you could be a bit more modest in the assumption relating linearity with high ice content (as you state at 1.457-462; 1.540-543). If not likely in these areas, it has to be stated.</li>
  - We acknowledge that not discussing rock slope instabilities and solifluction in our manuscript was a grave oversight and we adapted our manuscript to include them as potential causes of the observed

displacement. The landforms with a high linear velocities are mainly rock glaciers, protalus ramparts and collapsing moraines. Rock slope instabilities are present throughout the study sites but in those areas we observe mainly seasonally accelerated sliding.

d) Overall, maybe consider to use "downslope-dominated" (or gravity-driven) vs "verticallydominated" (or freeze/thaw-driven) instead of speaking about linearity/ seasonality. As you write at I.536-538 (too late and too little explained to my opinion), it can "have been misidentified as linearly moving, while actually featuring both the seasonal freeze-thaw cycle prevalent in the valleys and the

250 seasonal sliding pattern of the slopes". A way to check this it to plot the velocity in addition to (or instead of) the cumulated displacement on Fig.6. Fig.6B may look linear but looking at the velocity, I think

you may see variations as well.

245

- We changed Figure 6 of our manuscript to show a time series of the downslope velocity instead of cumulative surface displacement for B and C. We agree that it is important to acknowledge in our manuscript, that the landforms we describe as moving linearly also show variations in their velocities, albeit smaller than other landforms. We now also show the location of the landforms displayed in the time series.
- e) Information about several basic data properties and methodological information (important for 260 interpreting the results) is missing: multilooking factor, final spatial resolution, LOS angles, spatial/temporal baseline thresholds, temporal distribution of the initial SAR scenes (baseline plot), map with coherence, map with location of reference areas (in supplementary material for ex).
- We added information of the multi-looking factor, spatial resolution and incidence angles to the manuscript. We agree that supplementary material would be very helpful to assess the quality of our data. We therefore added a supplement which contains coherence maps including the locations of the reference points, maps of the interferogram percentage and baseline plots.
- f) About ISBAS processing: 1.189-190: "... where the coherence is intermittently below the chosen threshold..." and 1.198 "... near water bodies, where coherence is very low. We therefore decided to use a very low coherence threshold of 0.1 to increase...": I am not especially known with ISBAS approach, but this sounds quite dangerous to me, especially if you used a threshold of 0.1 in some areas (1.198). Does it mean that you have 25% of interferograms with <0.1 in these areas? Maximizing the coverage also to areas where the results cannot be reliable due e.g. to vegetation or moisture means that some of your interpretation can be based on wrong estimates. At least good to try to explain as much as possible the method, document the uncertainties and acknowledge the</li>
- potential limitations (in methods and/or in discussion). A coherence map could also be a nice way to document the distribution of these less reliable areas.

We expanded on the discussion regarding the limitations of our data and included coherence maps and interferogram percentage maps in the supplement.

g) Due to this lack of information, it is hard to fully understand the cause of the uplift detected in some flat valley bottoms (I.366 and Fig.3). Looking also at Fig.6A, if you subtract the last and the first acquisitions, you also get a positive trend. Is it really likely that all these locations are affected by sediment accumulation or can it be a bias? I wonder if this cannot be due to low reliability (low coherence) in these areas, especially during the "wet" periods when the ground is subsiding. Or a bias

285

stratified component)?

due to the temporal sampling of the initial interferograms? Or atmospheric effect (remaining

One major error source in periglacial environment (during summer) is the impact of ground moisture on the phase (moisture can lead to a biased detection of distance change, up to 10-20% of the wavelength). Good to discuss this. See e.g. references: De Zan et al., 2014; Zwieback et al., 2017.

We changed the paragraphs in question and discuss the possibility of a misinterpretation of soil moisture changes as uplift/sediment accumulation.

h) About DMS: at 1.413-414, it is written that there is shift of 11-27 days between ascending/ descending datasets. Why that? 11/27 days is quite a lot, considering that it should in theory document the same thing. Can it be due to a shift of velocity value (problem with the location of the reference points?) or the different LOS incidence angles (different sensitivity to the vertical)? Due to undocumented information about data properties, it is hard to understand the results and fully trust them.

We included information about the incidence angles and added a paragraph to the discussion section regarding the disparity between the DMS of ascending and descending data sets. We also included maps of this disparity in the supplement.

305

290

i) DMS 9 days prior to the temperature peak in one of the AOI is presented as "no lag" (I.422, I.427): this is in fact an inverse lag (or lag in the "wrong" direction), which has to be discussed. I would guess this may be due to the distance to the meteorological station: NAMORS station is maybe not representative of this AOI considering that Qugapie has a significantly higher elevation? Did you try to

- 310 apply an altitude correction? Figure 3: The bottom of the graph E/F is too little explained/exploited in the manuscript. To my opinion, this is maybe the most interesting finding of the study. There is a lack of structure in the Discussion. Consider dividing the Section in three parts, for ex: "Uncertainties/Error source"; "Thaw subsidence / Frost heave cycles"; "Downslope processes".
- 315

We reworked the discussion section to explore all mentioned points. We changed the structure to make it easier to follow.

- j) The introduction is overall a bit poor. It currently focuses a lot on the Tibetan Plateau, it could
   benefit for other references to similar kind of studies in others regions of the world. Here an nonexhausive list of ref. that could also contribute to go further with the discussion of your findings: in
   Alaska: Liu et al., 2010, 2012; Schaefer et al., 2015; in Canada: Short et al., 2014; Rudy et al., 2018; in
   Greenland: Strozzi et al., 2018; in Svalbard: Rouyet et al., 2019; in Siberia: Antanova et al., 2018.
- 325 We updated our manuscript to also present studies other regions besides the Tibetan Plateau in the introduction. We also compare results of these studies to our results in the discussion section.

k) Fig.3: Due to the chosen color scale, it is really hard to see the difference between areas with vertical assumption or those with downslope projection. Also hard to spot the areas affected by subsidence. + Maps C and D are only for Qugaqie basin. Why? + I may have missed sth, but I think there is no mention of the "seasonal sliding coefficient" threshold used to differentiate "linear velocity" and "faster in summer" in map C.

We changed the colour scales to differentiate better between vertical and downslope velocity. We added explanation of the seasonal sliding threshold to the relevant paragraphs and changed the legend of map 3C (now split into two figures and therefore called 7A) to make it clearer. The reason why the slopes of Niyaqu basin are not covered in such great detail compared to Qugaqie basin (not included in Figures 3 and 6 of our original manuscript) is that the spatial coverage of our InSAR data on the periglacial slopes of Niyaqu basin is much poorer. We added a paragraph to our manuscript to explain this. We included the map of the seasonal sliding model of Niyaqu basin in our supplement.

**365 InSAR time series analysis of seasonal surface displacement dynamics on the **Tibetan Plateau**

Eike Reinosch1, Johannes Buckel2, Jie Dong3, Markus Gerke1, Jussi Baade4, Björn Riedel1

[revised manuscript text omitted]

problem encountered by many studies investigating periglacial landscapes with InSAR techniques is heavy snow cover during the winter months (e.g. Eriksen et al., 2017), often leading to a complete loss of coherence. This is not a problem in our study sites at Lake Nam Co. In fact, we found that coherence is highest in winter, which we attribute to a stable frozen ground without growing vegetation.

This paper identifies the various surface displacement processes taking place around Nam Co on the southern TP and evaluates their potential causes. It is vital to understand these displacement 490 patterns and to compare our results to similar studies, as the TP reacts heterogeneously to climate change. Some lakes on the TP show a rising lake level, while others show stable or even falling lake levels (Mügler et al., 2010; Jiang et al., 2017). By assessing land surface displacements processes at Nam Co, we gain further information about the local situation, which allows us to set this region into accurate context compared to other regions of the TP.has been shown to react 495 heterogeneously to climate change (Song et al., 2014). To that end we developed multiple surface displacement models, analyzing geomorphological processes in the valleys and on the mountain slopes on both a seasonal and a multiannual scale. Furthermore we evaluate our hypothesis to predict if the creep of a periglacial landform is driven by its high ice content by differentiating between linear and seasonal motion patterns. This hypothesis is based on the assumption, that a 500 high ice content within the landform could facilitate significant creep throughout the year, leading to a linear motion pattern, while landforms without ice show no motion during periods when the ground is frozen, hence following a seasonal motion pattern.

**2 Study Area**

505 The Nam Co is the second largest lake of the TP (Zhou et al., 2013), with a catchment covering an area of 10,789 km2, 2018 km2 of which is the lake's own surface area (Zhang et al., 2017). The proximity to Lhasa, its accessibility and the presence of the scientific research station NAMORS (Fig.Nam Co Monitoring and Research Station for Multisphere Interactions CAS (NAMORS, Fig. 1), have made it a prime location to study the effects of climate change on the TP. The current lake level 510 lies at 4726 m a.s.l. (Jiang et al., 2017) but it has featured a rising trend of approximately 0.3 m yr 1over the past decades (Kropáček et al., 2012; Lei et al., 2013). <del>To the north and west the endorheic</del> catchment borders on the catchments of smaller lakes, such as Renco and Bamu Co.- The eastern and southern borders of the catchment are defined by the eastern and western Nyainqêntangha mountain ranges respectively. They feature range with elevations of up to 7162 m a.s.l. and are partially-The highest parts are glaciated (Bolch et al., 2010), while most other areas are considered to

515

---

## Author Response (AR2)

**Authors' response to referee comments:**

The referee's comments are shown in cursive with our responses below. We do not show every single comment made by the referee under 'other comments' but we made all the suggested changes of those comments that are not specifically mentioned here.

**Main comments:**

*1) Good that the temporal baselines are now documented (l.183-184). But it rises a new question: why the range is different for the two basins: 12-60 (Niyaqu) and 12-96 days (Ququaqie). Good to*
*quickly explain it. It is potentially a problem to compare the results cause the temporal baseline is related to the detection capability. Decorrelation can occur from 10.6 cm/yr for Ququaqie and 17 cm/yr for Niyaqu (and half of this for aliasing). You partly discuss it in 6.3 but good to also mention the max. velocities in 'Methods' and think about how it may have affected the comparison between the basins.*

We added this information to the relevant sections and mention it again in the discussion.

*2) The description of the different models is much easier to understand, but the table could still be improved to my opinion. Here some suggestions: in 'Purpose': for ex. LVM: Multi-annual subsidence, sediment accumulation and permafrost creep. HSM: Seasonal heave-subsidence cycle from AL*
*freezing and thawing. SSM: Seasonally accelerating slopes processes (as linear creep is included in LVM). In 'related geomorphological processes': LVM: add long-term subsidence and sediment accumulation, as you are also including <10 degrees areas. In 'associated landforms': In HSM: I guess this model is not only considering hummocks - maybe good to be a it more general. In SSM: debris mantle slopes, solifluction lobes, rockslides.*

We made all changes suggested by the referee.

*3) The discussion has been much improved. I think there are maybe still some elements missing related to the difference between ascending/descending results and the two basins: first, the temporal baseline is different (see comment 1). It can partly explain the difference in coherence and it*
*means that you may underestimate more high velocities in Ququaqie than in Niyaqu. Second, the time interval in Ququaqie is different for the 2 geometries (starting in June for ascending and November for descending). You have one more summer season in ascending geometry in Ququaqie. Third, the spatial distribution of the DMS difference: looking at S12/S13, it seems that the shift between ascending/descending is not randomly distributed, it has a spatial pattern. I have not thought much*
*more about the interpretation of these 3 elements and how this could influence your results, but I believe it should be considered.*

We expanded the section on these points in the discussion.

**Other comments:**

*l. 110: NAMORS, Fig. 1A*
*Fig. 1: You probably don't need to have information about data source both on the figure and in the*
*legend. I would remove the small texts in the figure: hard to read, redundant and so a bit useless.*

While we agree with the referee, we did not make the change they suggested, as it was specifically requested by the journal to have this information in both the figure itself and the figure text.

*l. 328-334: 'The larger the difference between the LOS vector and the vector representing the*
*assumed motion direction, in this case downslope, the smaller and therefore stronger the downslope*
*coefficient becomes': It is not the most intuitive sentence ever. And in general the whole explanation*
*about the coefficients is quite long and heavy to follow. Consider to rephrase and shorten. For ex. just*
*add a map with the coefficients in supplementary and refer to it without too much details.*

We shortened the text and included maps of the downslope coefficient in the supplement.

*Figure 3: As this combines two different projection according to the slope angle, it would be nice to*
*add in supplementary a slope map or a map with the classification </> 10 degrees to see where each*
*projection has been applied.*

We added the requested maps showing the classification of flat and steep terrain in both study areas to the supplement.

*l. 505-511: If it is not too much work, it would be nice to include in supplementary, one example for*
*each identified category (orthophoto or field pictures).*

We included field pictures of examples for the three landform types (rock glacier, moraine, rock slope instability) in the supplement.

*l. 622: '...where R2 >0.9 and rises to 29 where R2 <0.6'. How did you calculate sqrt R (in general, there*
*are several locations where you present error estimations and it is not clear how it has been found*
*out. 2-3 lines could be added in 'Methods').*

We added a short explanation in the methods section. We did not go into great detail regarding the calculation of $R^2$, as it is a widely used measure of for the quality of a regression curve.

[revised manuscript text omitted]
 along the slope or vertical | Seasonal vertical displacement | Seasonal displacement along the slope |
| **Slope** | <10°: vertical velocity >10°: along slope velocity | <10° | >10° |
| **Material** | Soil, regolith, till, debris and ice | Mainly soil | Regolith, debris and ice |
| **Related Geomorphological processes** | Long-term subsidence, sediment accumulation and pPermafrost creep | Heave-subsidence cycles connected to cryoturbation | Solifluction, gelifluction and rock slope instability on seasonally frozen slopes |
| **Associated Landform** | Rock glaciers, protalus ramparts and moraines | Hummocky Valley bottom terrain | Debris mantle slopes and lobates, solifluction lobes and rockslides |

**4.3.1 Linear Velocity Model (LVM)**

This model portrays the mean annual surface velocity, with different methods applied to regions with a slope >10° and with a slope <10°. Seasonal displacements trends are not present in this model, as
we address those in the separate models HSM and SSM. The original ISBAS processing chain (Sect. 4.1) is the same but we applied different methods to project the LOS results into a more meaningful direction. For areas with a slope <10° we assumed, that displacement would occur mainly in a vertical direction, as the slope would be too small to facilitate significant sliding or creep in most cases. Matsuoka (2001) shows that while solifluction has been documented on slopes as low as 2°,
most affected areas in mid-latitude to tropical mountains (including the TP) feature slopes >10°. To determine the vertical velocity, we performed a decomposition of ascending and descending time series data. For this process we assume the north-south component of the surface displacement to be zero, which allows us to determine the vertical and east-west components (Eriksen et al., 2017).

The vertical component represents our expected surface velocity for flat areas, while the east-west
component can be used to assess the error range of the velocity model.

The decomposition method works well for flat regions and slopes with an east or west aspect but
does not produce useful data for slopes with a north or south aspect. The Sentinel-1 SAR satellite
constellation is quite sensitive to both east-west and vertical surface displacement but very
insensitive to displacement with a strong north or south component. This is problematic when
studying displacements with a large horizontal component, as the velocity of surfaces moving in a
northern or southern direction will be either severely underestimated or completely overlooked. We
therefore employed a different method for slopes. Areas with a slope >10° were projected in the
direction of the steepest slope, as most surface displacement is assumed to be caused by sliding
processes transporting material parallel to the slope (Fig. 3). We made an exception for areas with an
east-west velocity >10 mm yr$^{-1}$, as our study areas feature a periglacial setting with landforms such as
rock glaciers, which move in a downslope direction and may extend into flatter areas. Those areas
were projected in a downslope direction, even on slopes <10°.  This approach (Notti et al., 2014)
originated from landslide studies, to produce a more accurate result for a process, where the
direction of the moving structure is either known or can be assumed with reasonable certainty. To
estimate the downslope velocity, we calculate a downslope coefficient, with values between 0.2 and
and divide the LOS velocity by this coefficient to determine the downslope velocity. Maps of the
spatial distribution of this coefficient are included in the supplement. , based on the LOS of the
satellite and the aspect and slope of the surface area. The larger the difference between the LOS
vector and the vector representing the assumed motion direction, in this case downslope, the
smaller and therefore stronger the downslope coefficient becomes. We excluded data points with a
strong coefficient if a slope has a strong coefficient in only one LOS but not the other, as a strong
coefficient is associated with a larger uncertainty. The maximum strength of the downslope
coefficient is set to 0.2 to avoid producing unrealistically large results caused by a coefficient close to

[revised manuscript text omitted]